# Autoregressive Generative Modeling with Noise Conditional Maximum Likelihood Estimation

## Abstract

We introduce a simple modification to the standard maximum likelihood estimation (MLE) framework. Rather than maximizing a single unconditional model likelihood, we maximize a family of *noise conditional* likelihoods consisting of the data perturbed by a continuum of noise levels. We find that models trained this way are more robust to noise, obtain higher test likelihoods, and generate higher quality images. They can also be sampled from via a novel score-based sampling scheme which combats the classical *covariate shift* problem that occurs during sample generation in autoregressive models. Applying this augmentation to autoregressive image models, we obtain 3.32 bits per dimension on the ImageNet 64x64 dataset, and substantially improve the quality of generated samples in terms of the Frechet Inception distance (FID) — from 37.50 to 12.09 on the CIFAR-10 dataset.

## 1 Introduction

Maximum likelihood estimation (MLE) is arguably the gold standard for probabilistic model fitting, and serves as the *de facto* method for parameter estimation in countless statistical problems Bishop (2006), across a variety of fields. Estimators obtained via MLE enjoy a number of theoretical guarantees, including consistency, efficiency, asymptotic normality, and equivariance to model reparameterizations Van der Vaart (2000). In the field of density estimation and generative modeling, MLE models have played a key role, where autoregressive models and normalizing flows have boosted competitive performance in a bevy of domains, including images Child et al. (2019), text Vaswani et al. (2017), audio Oord et al. (2016), and tabular data Papamakarios et al. (2017).

However, while log-likelihood is broadly agreed upon as one of the most rigorous metrics for goodness-of-fit in statistical and generative modeling, models with high likelihoods do not necessarily produce samples of high visual quality. This phenomenon has been discussed at length by Theis et al. (2015); Huszár (2015), and corroborated in empirical studies Grover et al. (2018); Kim et al. (2022). Autoregressive models suffer an additional affliction: they have notoriously unstable dynamics during sample generation Bengio et al. (2015); Lamb et al. (2016) due to their sequential sampling algorithm, which can cause errors to compound across time steps. Such errors cannot usually be corrected *ex post facto* due to the autoregressive structure of the model, and can substantially affect downstream steps as we find that model likelihoods are highly sensitive to even the most minor of perturbations.

Score-based diffusion models Song et al. (2020b); Ho et al. (2020) offer a different perspective on the data generation process. Even though sampling is also sequential, diffusion models are more robust to perturbations because, in essence, they are trained as denoising functions Ho et al. (2020). Moreover, the update direction in each step is unconstrained (unlike token-wise autoregressive models, which can only update one token at a time, and only once), meaning the model can correct errors from previous steps. However, likelihood evaluations have no closed form, requiring ODE/SDE solvers and hundreds to thousands of calls to the underlying network, and rendering the models incapable of being trained via MLE. Diffusion models also do not inherit any of the asymptotic guarantees Hyvärinen & Dayan (2005); Song et al. (2020a) of score matching, even though they

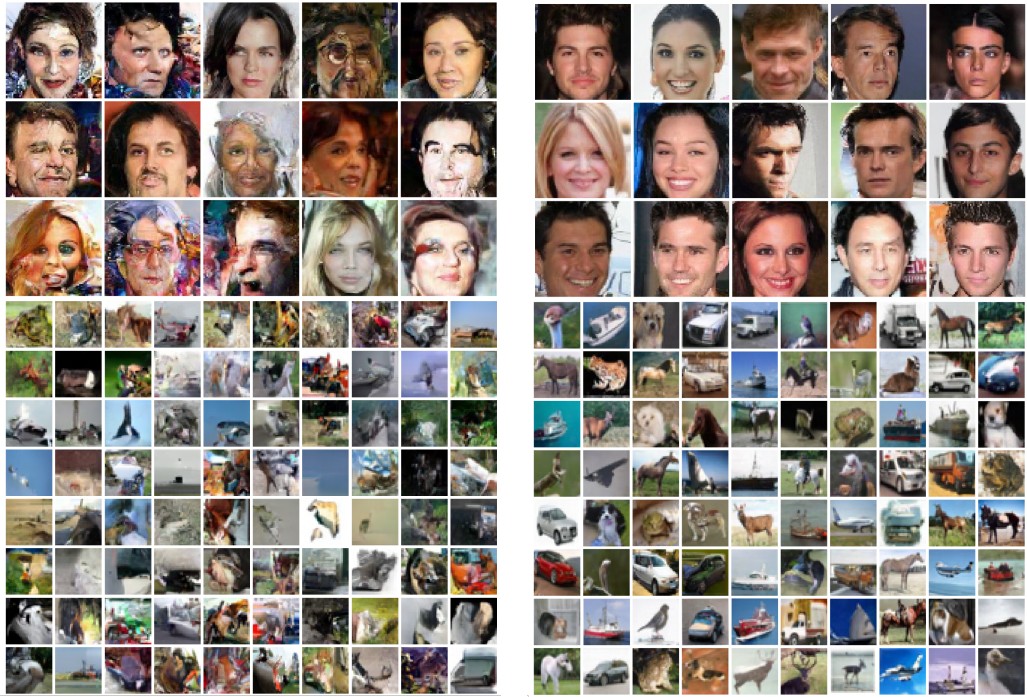

Figure 1: Generated samples on CelebA 64x64 **(above)** and CIFAR-10 **(below)**. Autoregressive models trained via vanilla maximum likelihood **(left)** are brittle to sampling errors and can quickly diverge, producing nonsensical results. Those trained by our algorithm **(right)** are more robust and ultimately generate more coherent sequences.

are thusly trained [1]. These details make diffusion models theoretically inferior and less viable for many downstream tasks that involve the likelihood, such as anomaly and out-of-distribution (OOD) detection Ren et al. (2019), adversarial defense Song et al. (2017), among others. Thus we wonder: is there a conceptual middle ground?

In this paper, we offer such a framework. We further analyze the likelihood-sample quality mismatch in autoregressive models, and propose techniques inspired by diffusion models to alleviate it. In particular, we leverage the fact that the score function is naturally learned as a byproduct of maximum likelihood estimation. This allows a novel two-part sampling strategy with noisy sampling and score-based refinement.

Our contributions are threefold. 1) We investigate the pitfalls of training and inference under the maximum likelihood estimation framework, particularly regarding sensitivity to noise corruptions. 2) We propose a simple sanity test for checking the robustness of likelihood models to minor perturbations, and find that many models fail this test. 3) We introduce a novel framework for the training and sampling of MLE models that significantly improves the noise-robustness and generated sample quality. As a result, we obtain a model that can generate samples at a quality approaching that of diffusion models, without losing the maximum likelihood framework and $\mathcal{O}(1)$ likelihood evaluation speed of MLE models.

## 2 BACKGROUND AND RELATED WORK

Let our dataset $\mathcal{X}$ consist of i.i.d. samples drawn from an unknown target density $\mathbf{x} \sim p_{data}(\mathbf{x})$. The goal of likelihood-based generative modeling is to approximate $p_{data}$ via a parametric model $p_{\boldsymbol{\theta}}$, where samples $x \sim p_{\boldsymbol{\theta}}$ can be easily obtained.

---

[1]Though each conditional score is trained via score matching, the final model depends heavily on the solution of a chosen SDE which is not specified by the framework. Thus score matching does not directly produce a diffusion model.

## 2.1 BACKGROUND

We first discuss fundamental techniques for estimating and sampling from $p_\theta$ in generative modeling.

**Score-based Modeling** A technique for generative modeling that has recently grown in popularity is score-based modeling, which involves minimizing a sum of weighted score matching losses Hyvärinen & Dayan (2005) on noise-corrupted data

$$\arg\min_{p_{\boldsymbol{\theta},t}} \int_t \mathbb{E}_{\mathbf{x}\sim p_{data}} \mathbb{E}_{\tilde{\mathbf{x}}\sim p(\tilde{\mathbf{x}}|\mathbf{x})} ||\nabla_{\mathbf{x}} \log p_{\boldsymbol{\theta},t}(\tilde{\mathbf{x}}) - \nabla_{\mathbf{x}} \log p_{data}(\tilde{\mathbf{x}})||^2 \mu(t)dt, \tag{1}$$

where $p(\tilde{\mathbf{x}}|\mathbf{x})$ describes a corruption process and $\nabla_{\mathbf{x}} \log p(\mathbf{x})$ is also known as the *(Stein) score* function. Sampling from $p_\theta$ can then be achieved via annealed Langevin dynamics Song & Ermon (2019), variational denoising Ho et al. (2020), or reversing a diffusion process Song et al. (2020b).

To obtain a proper likelihood value, score-based models must be framed as a diffusion process, where the density is then obtained as the solution of a related ordinary differential equation (ODE). Here, each data point is modeled as a function $\mathbf{x} : [0, T] \rightarrow \mathbb{R}^d$ such that $\mathbf{x}(0) \sim p_{data}$ and $\mathbf{x}(T) \sim p_{prior}$. The forward diffusion process is an Ito stochastic differential equation (SDE)

$$d\mathbf{x} = \mathbf{f}(\mathbf{x}, t) + g(t)d\mathbf{w}, \tag{2}$$

for some drift and diffusion terms $\mathbf{f}$ and $g$, where $\mathbf{w}$ is the standard Wiener process. By a result in Anderson (1982), this diffusion can be tractably reversed

$$d\mathbf{x} = [\mathbf{f}(\mathbf{x}, t) + g^2(t) + \nabla_{\mathbf{x}} \log p_t(\mathbf{x})]dt + g(t)d\bar{\mathbf{w}}, \tag{3}$$

and depends additionally on the noise-conditional score function, where $\bar{\mathbf{w}}$ is a backwards Wiener process. Sampling then consists of drawing $\mathbf{x}(T) \sim p_{prior}$, and solving the reverse diffusion process.

**Maximum Likelihood Estimation (MLE)** The standard MLE framework consists in solving

$$\arg\max_{p_{\boldsymbol{\theta}}} \mathbb{E}_{\mathbf{x}\sim p_{data}} \log p_{\boldsymbol{\theta}}(\mathbf{x}) \approx \arg\max_{p_{\boldsymbol{\theta}}} \frac{1}{|\mathcal{X}|} \sum_{\mathbf{x}\in\mathcal{X}} \log p_{\boldsymbol{\theta}}(\mathbf{x}). \tag{4}$$

Likelihood models draw samples $\mathbf{x} \sim p_\theta$ one of two ways. Normalizing flows Dinh et al. (2014); Rezende & Mohamed (2015); Grathwohl et al. (2018) apply a series of invertible transformations to a latent variable $\mathbf{z} \in \mathbb{R}^d \sim p_{prior}$. On the other hand, autoregressive models sample $\mathbf{x}$ sequentially and coordinate-wise by drawing from each conditional likelihood.

In this work, we draw an explicit distinction between MLE and other parameter estimation techniques that increase the likelihood, such as score-based modeling or variational lower bound maximization, which is sometimes regarded as maximum likelihood Song et al. (2021). The key difference is that MLE provides theoretical guarantees on the resulting model — including consistency, efficiency, asymptotic normality Van der Vaart (2000) — whereas score-based modeling and variational lower bound maximization do not.

## 2.2 RELATED WORK

A number of other works combine score- and energy-based modeling with autoregressive architectures. Hoogeboom et al. (2021) propose an order-agnostic autoregressive model for simulating *discrete* diffusions via a variational lower bound. Meng et al. (2020) use unnormalized autoregressive models to learn distributions in an augmented score-matching framework. Nash & Durkan (2019) design an energy-based model with an autoregressive structure such that the normalizing constant can be estimated coordinate-wise via importance sampling. Similar to our approach, Meng et al. (2021) decompose the data generation process into a noisy sampling step and a denoising step, but their formulation introduces latent variables and does not support adaptive refinement strategies. Ultimately, each approach relinquishes the ability to compute exact likelihoods in their framework, which is one of the motivating advantages of using autoregressive structures.

A closely related vein of research explores alternative training and inference strategies so as to improve sampling stability in autoregressive models. Bengio et al. (2015) propose training models

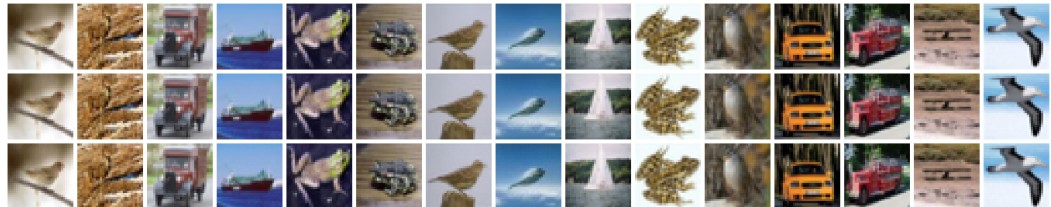

Figure 2: Images from CIFAR-10 **(top)** versus their $p_\pi$ perturbed counterparts for $\pi = 0.5$ **(middle)** and $\pi = 1$ **(bottom)**. The differences are nearly indistinguishable to the human eye, yet cause drastic deviations in average log-likelihood for standard likelihood models (Section 3 and Table 1).

with a mixture of true and generated samples over time, where the proportion of generated samples gradually grows to take up the majority of training sequences. This is initially promising, except Huszár (2015) note that this technique is biased and not guaranteed to produce solutions that converge on the true distribution. Lamb et al. (2016) subsequently suggest incorporating an adversarial loss provided by a discriminator that "teaches" the model to produce more realistic samples over multiple sampling steps. Both techniques again depart from the maximum likelihood framework. Jun et al. (2020) considers alternative forms of data augmentation, which experimentation in Kingma et al. (2021) show is synergistic with noise conditional perturbations during training. Perhaps most similar to our approach is Jayaram & Thickstun (2021), who also suggest sampling via the score function with Langevin dynamics, but they crucially do not train with noise, which Song & Ermon (2019) found to be essential for stable sampling in a Langevin algorithm. As a result, they are not able to sample images unconditionally.

Ultimately, our approach uniquely provides a principled and generalized framework for modeling stochastic processes (including diffusions) that retains the asymptotic guarantees of maximum likelihood estimation while producing samples of superior quality.

## 3 THE PITFALLS OF MAXIMUM LIKELIHOOD ESTIMATION

We first show that density models trained to maximize the standard log-likelihood are surprisingly sensitive to minor perturbations. We then discuss why this is bad for generative modeling performance.

### 3.1 A SIMPLE SANITY TEST

Consider the class of minimally corrupted probability densities on a discrete $k$-bit integer space (*e.g.*, 8-bit images, or 16-bit digital audio signals):

$$p_\pi(\tilde{\mathbf{x}}) = \sum_{\mathbf{x}=1}^{2^k} q_\pi(\tilde{\mathbf{x}}|\mathbf{x})p(\mathbf{x}), \qquad q(\tilde{\mathbf{x}}|\mathbf{x}) = \begin{cases} \mathbf{x} & \text{w.p. } 1-\pi \\ \texttt{bitflip}(\mathbf{x}, k) & \text{w.p. } \pi \end{cases}, \tag{5}$$

where $\pi \in [0, 1]$ and $\texttt{bitflip}(\mathbf{x}, \ell)$ is the bit flip operation on the $\ell$th bit of $\mathbf{x}$. $p_\pi$ is *minimally corrupted* in the sense that $p_\pi$ describes the distribution of points in $p_{data}$ that have been perturbed by flipping the least significant bit (LSB) with probability $\pi$.

In 8-bit images, the difference between samples drawn from $p_\pi$ and $p_{data}$ is imperceptible to the human eye, even for $\pi = 1$ (see Fig 2). However, for likelihood models, this perturbation drastically increases the negative log-likelihood (Table 1). In the next section, we provide three reasons for why failing this test is problematic, especially for autoregressive models.

### 3.2 WHAT THIS MEANS (FOR GENERATIVE MODELING AND DENSITY ESTIMATION)

First, noise is natural — and being less robust to noise also means being a poorer fit to natural data, especially in the ways that matter to the end user. Outside of the log-likelihood, measures of generative success in generative models fall under two categories: qualitative assessments (*e.g.*,

the no-reference perceptual quality assessment Wang et al. (2002) or 'eyeballing' it) and quantitative heuristics (*e.g.*, computing statistics of hidden activations of pretrained CNNs Salimans et al. (2016); Heusel et al. (2017); Sajjadi et al. (2018)). Both strategies either rely directly on the human visual system, or are known to be closely related to it Güçlü & van Gerven (2015); Yamins et al. (2014); Khaligh-Razavi & Kriegeskorte (2014); Eickenberg et al. (2017); Cichy et al. (2016). Therefore, implicit in the use of these criteria is the existence of a human (or human-like) model of images $q_{human}$, where $q_{human} \approx p_{data}$ Huszár (2015). Our sanity test thus shows that we are still quite far from obtaining $q_{human}$. We summarize this observation with the following lemma.

**Lemma 3.1.** *(Informal) Let $q_{human}$ be a human-like distribution on images, and $p_\theta$ be a likelihood model that is not robust to a visually imperceptible perturbation $q_\pi$. Then $p_\theta$ cannot be equal to $q_{human}$ almost surely.*

Second, generative sample quality suffers, especially for autoregressive models. Due to the sequential nature of autoregressive sampling and the fact that models are presented only with data from the *true* distribution during training, autoregressive models are already known to be poorly-equipped to handle the sequences they encounter during sampling Bengio et al. (2015); Lamb et al. (2016); Meng et al. (2021). Any sampling error introduces a distributional shift that can affect the modeled distribution on downstream tokens. This will increase the risk of mis-sampling the next token, which, in turn, further affects downstream modeling. This is related to the well-known *covariate shift* phenomenon Shimodaira (2000). Sensitivity to minor perturbations only exacerbates the problem, and Table 1 shows that in vanilla autoregressive likelihood models, mis-sampling pixels by even a single bit can cause drastic changes to the overall likelihood. This can explain why such models commonly produce nonsensical results (Fig 1).

Third, we may need to re-evaluate our research trajectory in image-based density estimation. One way to interpret Table 1 is via the following lemma.

**Lemma 3.2.** *Let $q$ be a distribution on $d$-dimensional $k$-bit data $\mathbf{x} \in \{0, \ldots, 2^k\}^d$. Denote $\mathbf{x}_i \in \{0, 1\}^d$ as the bit mask containing the $i$th bit of $\mathbf{x}$. When $\pi = \frac{1}{2}$, we have*

$$\mathbb{E}_{\tilde{\mathbf{x}} \sim p_\pi}[q(\tilde{\mathbf{x}})] = \mathbb{E}_{\mathbf{x} \sim p_{data}}[q(\mathbf{x}_1, \ldots, \mathbf{x}_{k-1})], \tag{6}$$

*where $q(\mathbf{x}_1, \ldots, \mathbf{x}_{k-1}) := \int_{\mathbf{x}_k} q(\mathbf{x})dx$ is the marginal of $q(\mathbf{x})$ (after marginalizing out $\mathbf{x}_k$).*

In other words, the expected likelihood with the full $k$-bit image over $p_{\frac{1}{2}}$ is equivalent to the expected likelihood with the first $k-1$ most significant bits over $p_{data}$. Table 1 shows that while there is a large variance in likelihoods with $\pi = 0$, most models achieve between $3.75 - 3.85$ BPD with $\pi = 0.5$. This suggests that most gains so far in likelihood modeling have been focused on modeling the least significant bit, which ultimately bears little significance to the inherent content of the image. Conversely, NCML trained models are significantly more effective at modeling the remaining bits.

For these reasons, we find that improving noise-robustness is of central importance to likelihood-based generative modeling, and especially likelihood-based autoregressive generative models.

## 4 NOISE CONDITIONAL MAXIMUM LIKELIHOOD

To alleviate the problems discussed in Section 3, we propose a simple modification to the standard objective in maximum likelihood estimation. Rather than evaluating a single likelihood as in the vanilla formulation, we consider a family of noise conditional likelihoods

$$\mathbb{E}_{t \sim \mu} \mathbb{E}_{\mathbf{x} \sim p_t} \log p_{\boldsymbol{\theta}, t}(\mathbf{x}), \tag{7}$$

where $p_t$ is a stochastic process indexed by noise scales $t$ modeling a noise-corrupting process on $p_{data}$, and $\mu$ is a distribution over such scales. We call this the **noise conditional maximum likelihood (NCML)** framework. In general, Eq 7 is an all-purpose plug-in objective that can be used with any likelihood model adapted to accept a noise conditioning vector[2]. We now explore various perspectives of NCML that may help with reasoning about this framework.

---

[2]Though a continuous likelihood or a discretization of it (e.g. normalizing flows Dinh et al. (2014) or autoregressive models with non-softmax probabilities Salimans et al. (2017); Li & Kluger (2022)) is necessary for computation of the score function.

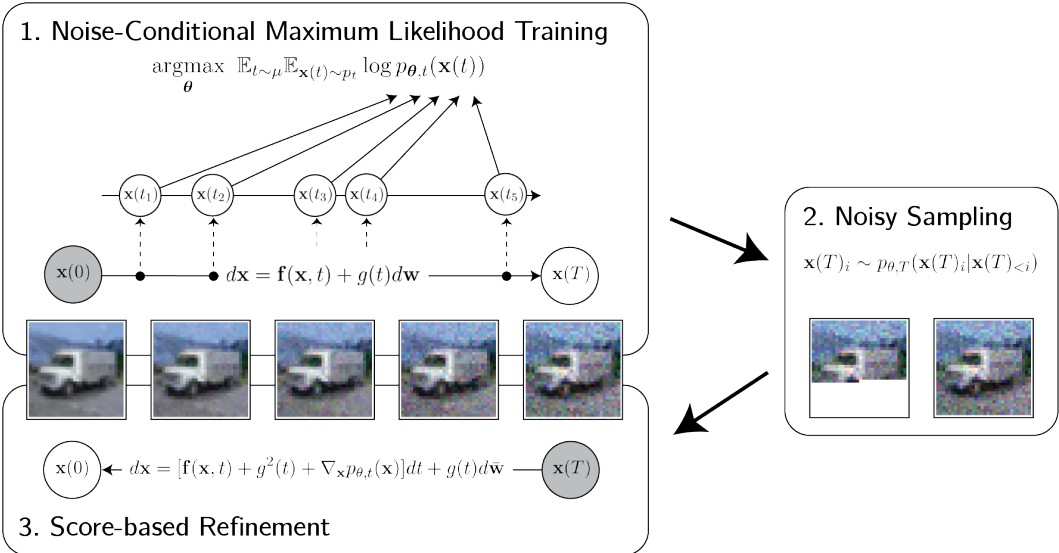

Figure 3: An overview of the NCML generative algorithm. There are three steps: 1) We train the noise conditional density model $p_{\theta,t}$ via NCML. 2) We sample from the modeled noisy distribution, *i.e.*, $p_{\theta,t}$ with some $t > 0$. 3) We refine the sample by solving a reverse diffusion involving the learned score function $\nabla_x p_{\theta,t}(x)$.

**NCML as Regularized Maximum Likelihood** When the set of noise scales $t \in \mathcal{T}$ is finitely large, a natural and mathematically equivalent formulation of NCML is as a form of data- and model-dependent regularization of the standard maximum likelihood estimation objective:

$$\mathbb{E}_{t\sim\mu}\mathbb{E}_{\mathbf{x}\sim p_t} \log p_{\boldsymbol{\theta},t}(\mathbf{x}) \propto \underbrace{\mathbb{E}_{\mathbf{x}\sim p_{data}} \log p_{\boldsymbol{\theta},0}(\mathbf{x})}_{\text{MLE objective}} + \underbrace{\sum_{t\in\mathcal{T}} \lambda_t \mathbb{E}_{\mathbf{x}\sim p_t} \log p_{\boldsymbol{\theta},t}(\mathbf{x})}_{\text{regularization term}}, \quad (8)$$

where $\mathcal{T}$ comprises the set of nonzero noise scales and $\lambda_t := \mu(t)/\mu(0)$. Clearly, the standard likelihood can be considered a special case of our proposed method where $\lambda_t = 0$ for all $t \in \mathcal{T}$. Furthermore, since the NCML framework can simply be seen as formulating $|\mathcal{T}|$ simultaneous and separate MLE problems, it retains all the statistical properties of standard MLE.

Of course, any form of regularization introduces bias to the model framework. Whereas L0/L1/L2 regularizations bias towards solutions of minimal or sparse weight norm, our experiments suggest that NCML biases towards solutions that are less sensitive to noisy perturbations (see Section 4.1).

**NCML as a Diffusion Model** Letting $t$ be the time index of a diffusion process, our approach becomes closely related to score-based diffusion models Song et al. (2020b), albeit with two crucial differences.

First, instead of merely estimating the noise-conditional score $\nabla_x \log p_t(x)$ for $t \in \mathcal{T}$, we directly estimate $p_t$ itself. However, $\nabla_x \log p_t(x)$ is still learned as a by-product of NCML, as it is the derivative of the log of the learned quantity. We may then access our approximated score via standard backpropagation techniques. Therefore, like diffusion models, we can draw samples via Langevin dynamics. This provides an alternative strategy for sampling from $p_{\theta,t}$, which we explore in 4.2.

Second, we need not design our diffusion so that $p_T$ approximates the limiting stationary distribution of the process. This is necessary in diffusion models as the limiting prior is the only tractable distribution to initialize the sampling algorithm with. Since we have learned the density itself for all $t \in \mathcal{T}$, we may initialize from any point of the diffusion, which increases the flexibility of the sampling strategy, and can drastically reduce the steps required to solve the reverse diffusion.

For our models, we consider the three diffusion processes proposed in Song et al. (2020b): variance exploding (VE), variance preserving (VP), or sub-variance preserving (sub-VP), and choose $\mu$ to be

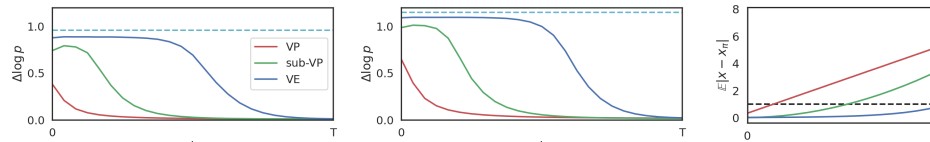

Figure 4: Noise robustness of NCML-trained models with Variance Preserving (VP), sub- Variance Preserving (sub-VP), and Variance Exploding (VE) noise schedules, measured in terms of $\Delta \log p$ (defined in 9) between $p_{data}$ and $p_\pi$, for $\pi = 0.5$ **(left)** and $\pi = 1$ **(middle)**. **(Right)** shows that robustness is closely related to the average absolute perturbation per pixel of each noise schedule, as a function of $t$. More details in Section 4.1.

the uniform distribution over $\mathcal{T}$. Due to space constraints, we refer to the aforementioned paper for more details on these SDEs.

## 4.1 NOISE-ROBUSTNESS OF NCML MODELS

We find that models trained via NCML are more robust to noise, suggesting increased stability during autoregressive sampling and improved modeling of more visually salient features (Section 3.2). Surprisingly, this is not only the case for noise conditions $t > 0$, where the model is exposed to noisy samples. Indeed, even in the $t = 0$ condition, noise-robustness is improved (Table 1), and our models surpass state-of-the-art likelihood models on minimally perturbed data (as defined in Eq 5) in terms of average log-likelihood. This is somewhat unexpected: by passing the noise condition, the model should theoretically be able to separate the NCML loss into distinct problems at each noise scale. If this occurs, then the $t = 0$ case should simplify to a vanilla MLE problem, where behavior would not differ from standard likelihood models.

This lends credence to the regularization perspective provided in 4. In essence, noise robustness is explicitly enforced for $t > 0$. For $t = 0$, NCML leverages the limited capacity of the underlying network to implicitly impose robustness. To quantify the noise-robustness at each $t$, we define the simple measure $\Delta \log p$ as the absolute difference between the negative model log likelihood (as measured in bits per dimension) evaluated on $p_{data}$ minus that on $p_\pi$, *i.e.*,

$$\Delta \log p := |\mathbb{E}_{p_{data}} \log p_{\boldsymbol{\theta}} - \mathbb{E}_{p_\pi} \log p_{\boldsymbol{\theta}}|. \tag{9}$$

The left and middle graphs in Fig 4 show $\Delta \log p$ as a function of $t$ for $\pi = 0.5$ and $\pi = 1$, respectively. Here, we can clearly see that noise robustness increases with increasing $t$. Moreover, the regularization effect of NCML enforces greater robustness than competing models even at $t = 0$, as seen by the dotted line showing the next lowest $\Delta \log p$, indicating that NCML enforces some degree of noise-robustness as regularization.

As expected, the correlation between noise-robustness and $t$ follows closely to the noise schedules of VP, sub-VP, and VE SDEs in $[0, T]$ respectively. This can be seen in the rightmost plot of Figure 4, which shows the average absolute perturbation per pixel of each SDE over time. The dotted line represents 1, *i.e.*, the absolute perturbation of corrupted images in our sanity test $p_\pi$. The point at which each model attains robustness to $p_\pi$ corrupted noise is more or less the same time the noise schedule begins to perturb each pixel by at least one bit, on average.

The increased noise-robustness of NCML-trained models at larger $t$ motivates our improved autoregressive sampling algorithm, which we introduce below.

## 4.2 SAMPLING WITH AUTOREGRESSIVE NCML MODELS

Our framework allows for two sampling strategies. The first is to draw directly from the noise-free distribution $p_{\boldsymbol{\theta},0}$, in which case the conditional likelihood simplifies to a standard (unconditional) likelihood, and sampling is identical to that for a vanilla autoregressive model.

However, as discussed in Section 3, this strategy is unstable and tends to quickly accumulate errors. This motivates an alternative two-part sampling strategy, which involves drawing from $p_{\theta,t}$ for $t > 0$ (the *noisy sampling* phase), then solving a reverse diffusion process back to $t = 0$ (the *score-based*

| Model | | CIFAR-10 | | | ImageNet 64x64 | | |
|---|---|---|---|---|---|---|---|
| | FID | NLL $\pi = 0$* | NLL $\pi = 0.5$ | NLL $\pi = 1$ | NLL $\pi = 0$* | NLL $\pi = 0.5$ | NLL $\pi = 1$ |
| **ELBO** | | | | | | | |
| VDM | 7.41 | **2.49** | **3.76** | **3.96** | **3.40** | 3.76 | 3.87 |
| ScoreFlow | **5.40** | 2.90 | 3.83 | 3.98 | - | - | - |
| VDVAE | - | 2.84 | 3.89 | 4.10 | 3.52 | **3.63** | **3.82** |
| **Likelihood** | | | | | | | |
| Flow++ | - | 3.09 | 3.84 | 4.09 | 3.69 | 3.82 | 3.99 |
| DenseFlow | 48.15 | 2.98 | 3.79 | 4.02 | 3.35 | 3.69 | 3.85 |
| PixelCNN++ | 55.72 | 2.92 | 3.85 | 4.02 | 3.52 | 3.81 | 3.99 |
| PixelSNAIL | 36.62 | 2.85 | 3.83 | 3.99 | - | - | - |
| Sparse Transformer | 37.50 | **2.80** | 3.82 | 3.97 | 3.44 | 3.73 | 3.89 |
| NCPN (ML) | 46.72 | 2.91 | 3.82 | 3.99 | 3.49 | 3.68 | 3.90 |
| NCPN (NCML-VE) | 32.71 | 2.87 | 3.73 | 3.94 | **3.32** | 3.67 | 3.85 |
| NCPN (NCML-subVP) | 23.42 | 2.95 | 3.68 | 3.93 | 3.36 | 3.66 | 3.80 |
| NCPN (NCML-VP) | **12.09** | 3.20 | **3.61** | **3.89** | 3.43 | **3.62** | **3.76** |

Table 1: Results on CIFAR-10 and ImageNet 64x64. Negative log-likelihood (NLL) is in bits per dimension. Lower is better. *NLL with $\pi = 0$ is equivalent to NLL of the original data.

*refinement* phase). The tractability of the latter is due to the fact that NCML-trained models learn the score as a byproduct of likelihood estimation. This is identical to the sampling procedure in score-based diffusion models Song et al. (2020b), except for the key difference that we need not initialize with the prior distribution, as we can sample from any $t \in \mathcal{T}$.

This two-part strategy has several benefits. First, we saw in Section 4.1 that NCML-trained models are more robust to noise at higher $t$, which improves stability during sampling. This is reflected in the improved FID of NCML-trained NCPNs compared to ML- (*i.e.*, maximum likelihood-) trained NCPNs in Table 1. Second, the refinement phase allows the model to make fine-tuned adjustments to the sample, which can further improve quality. This is not possible in standard autoregressive models by construction.

## 5 EXPERIMENTS

We demonstrate that incorporating noise in the maximum likelihood framework provides significant improvements in terms of density estimation, sample generation, and anomaly detection. In all experiments, we fix $p_t$ to be one of the variance exploding (VE), variance preserving (VP), or sub-variance preserving (sub-VP) SDEs, and $\mu$ to be the uniform distribution over $t \in \mathcal{T}$. For our architecture, we introduce the noise conditional pixel network (NCPN), which consists of a Pixel-CNN++ Salimans et al. (2017) backbone with added attention layers. More experimental details can be found in A.2.

**Unconditional Modeling** We evaluate our models on minimally perturbed transformations (see Section 3.1) of unconditional CIFAR-10 and ImageNet 64x64 for $\pi \in \{0, \frac{1}{2}, 1\}$, where we note that $p_\pi = p_{data}$ when $\pi = 0$. All noise conditional models, *i.e.*, ours, VDM Kingma et al. (2021), and ScoreFlow Song et al. (2021), are evaluated at $t = 0$. We show our results in Table 1. We additionally evaluate our model on the CelebA 64x64 dataset; autoregressive comparisons on this dataset are limited, so we defer results to the appendix.

On the standard unperturbed dataset $\pi = 0$, our models attain competitive likelihoods on CIFAR-10 and state-of-the-art likelihoods on ImageNet 64x64. On all perturbed datasets, our models achieve state-of-the-art likelihoods. Furthermore, we significantly improve on the state-of-the-art in terms of sample quality among MLE models on CIFAR-10, from 37.50 to 12.09 in terms of FID. In general, our results indicate that the NCML framework improves generative modeling performance of the underlying models in terms of both test log-likelihoods and sample quality.

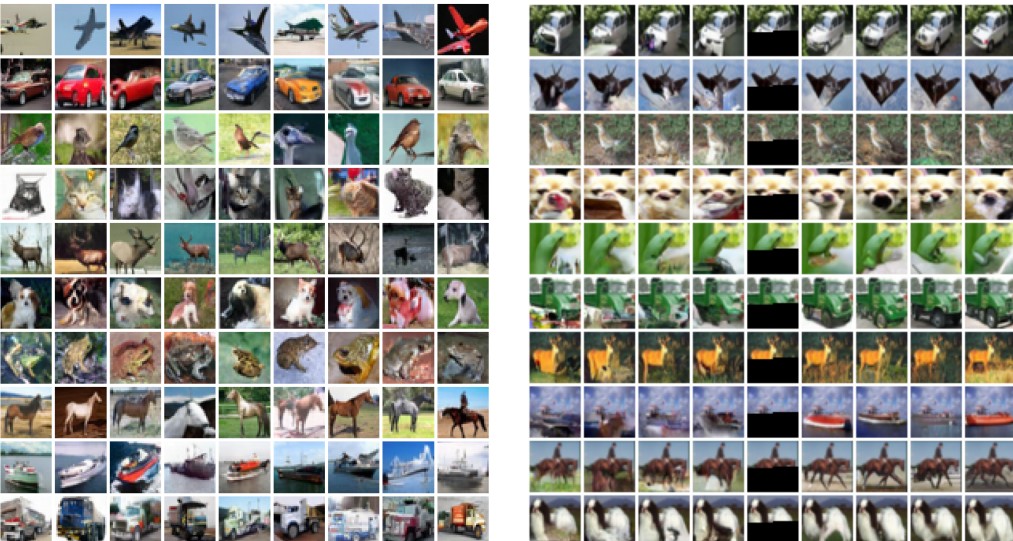

Figure 5: Class-conditional sampling on CIFAR-10 **(left)**. Image completion on CIFAR-10 **(right)**. See Section 5 for more details.

**Class-conditional Modeling** We find that NCML-trained models exhibit stable sampling even in class-conditional generation. For this experiment, we train a class-conditional model under our framework on the CIFAR-10 dataset. We show our results in Figure 5, where each row shows data sampled from a different class of CIFAR-10.

**Image Completion** We further examine NCML-trained models in a controllable generation context through the image completion task, which involves conditioning an autoregressive model on the first half of an image, and drawing the second half from the modeled conditional distribution. We again use the CIFAR-10 dataset, and compare models trained under our framework against those trained with MLE. All images are taken from the test set to minimize data memorization. Our results are in figure 5 (right). The leftmost and rightmost four columns are outputs generated by an MLE-trained model and an NCML-trained model, respectively, with the middle column depicting the masked input to the sampling algorithm. Both models use the same architecture. Our model demonstrates improved stability over the course of sampling, and produces completed images with greater realism and fidelity, while maintaining a high diversity of sampled trajectories.

**Out-of-distribution (OOD) Detection** We show that robustness to minimally perturbed data confers benefits that extend beyond enhanced sample quality, including marked improvements in anomaly detection. In this task, the network is trained on in-distribution points, then tasked to produce a statistic that successfully differentiates in-distribution points from out-of-distribution points. Following Du & Mordatch (2019); Meng et al. (2020), we train models on CIFAR-10, and let the OOD points be SVHN Netzer et al. (2011), constant uniform, and random uniform images. We compare the Area Under the Receiving Operator Curve (AUROC) of our model statistic to vanilla PixelCNN++ Salimans et al. (2017), GLOW Kingma & Dhariwal (2018), EBM Du & Mordatch (2019), and AR-CSM Meng et al. (2020) models, and see that our model performs equally or better in every respect (Table 2).

| Data/Model | PixelCNN++ | GLOW | EBM | AR-CSM | NCPN (VP) |
|---|---|---|---|---|---|
| SVHN | 0.32 | 0.24 | 0.63 | 0.68 | **0.74** |
| Const Uniform | 0.0 | 0.0 | 0.30 | 0.57 | **0.68** |
| Uniform | **1.0** | **1.0** | **1.0** | 0.95 | **1.0** |
| Average | 0.44 | 0.41 | 0.64 | 0.73 | **0.80** |

Table 2: AUROC scores of models trained on CIFAR-10 on the OOD detection task.

## 6 CONCLUSION AND FURTHER WORK

We proposed a simple sanity test for checking the robustness of likelihoods to visually imperceptible levels of noise, and found that most models are highly sensitive to perturbations under this test. We argue that this is further evidence of a fundamental disconnect between likelihoods and other sample quality metrics. To alleviate this issue, we developed a novel framework for training likelihood models that combines autoregressive and diffusion models in a principled manner. Finally, we find that models trained under this setting have substantial improvements in both training and evaluation.

While models trained under the NCML framework show greater invariance to imperceptible noise, they are by no means robust, indicating that the underlying model still differs significantly from the theoretical human model $q_{human}$ proposed in Huszár (2015). We hope that further research can help close this gap, and furnish us with a more intuitive grasp on the maximum likelihood as a framework for assessing goodness-of-fit in generative models.

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

## A APPENDIX

### A.1 PROOFS

**Lemma 3.1.** *(Formal) Let $q_{human}$ and $p_\theta$ be continuous and bounded distributions supported on $\mathbb{R}^d$. We assume that $q_{human}$ is human-like. Therefore, if there exists a perturbation $q_\pi$ such that samples drawn from $p_{data}$ and $p_\pi$ (as defined in 2) are visually indistinguishable to the human eye, then $E_{\mathbf{x} \sim p_\pi}[\log q_{human}(\mathbf{x})] = E_{\mathbf{x} \sim p_{data}}[\log q_{human}(\mathbf{x})]$. Further assume that $E_{\mathbf{x} \sim p_\pi}[\log p_\theta(\mathbf{x})] = E_{\mathbf{x} \sim p_{data}}[\log p_\theta(\mathbf{x})]$. Then $p_\theta$ cannot be equal to $q_{human}$ almost surely.*

*Proof.* We show this by contradiction. Suppose that $q_{human} = p_\theta$ almost everywhere. Then for any distribution $\nu$,
$$E_{\mathbf{x} \sim \nu}[p_\theta(\mathbf{x})] = E_{\mathbf{x} \sim \nu}[q_{human}(\mathbf{x})].$$

Let $p_{data}$ and $p_\pi$ be two distributions, where by assumption
$$E_{\mathbf{x} \sim p_{data}}[q_{human}(\mathbf{x})] = E_{\mathbf{x} \sim p_\pi}[q_{human}(\mathbf{x})],$$

while
$$E_{\mathbf{x} \sim p_{data}}[p_\theta(\mathbf{x})] \neq E_{\mathbf{x} \sim p_\pi}[p_\theta(\mathbf{x})].$$

Thus we have
$$E_{\mathbf{x} \sim p_{data}}[p_\theta(\mathbf{x})] = E_{\mathbf{x} \sim p_{data}}[q_{human}(\mathbf{x})] = E_{\mathbf{x} \sim p_\pi}[q_{human}(\mathbf{x})] = E_{\mathbf{x} \sim p_\pi}[p_\theta(\mathbf{x})], \qquad (10)$$

but we assumed that $E_{\mathbf{x} \sim p_{data}}[p_\theta(\mathbf{x})] \neq E_{\mathbf{x} \sim p_\pi}[p_\theta(\mathbf{x})]$, which is a contradiction. Thus we have the desired result. $\qquad\square$

**Lemma 3.2.** *Let $q$ be a distribution on $d$-dimensional $k$-bit data $\mathbf{x} \in \{0, \ldots, 2^k\}^d$. Denote $\mathbf{x}_i \in \{0, 1\}^d$ as the bit mask containing the $i$th bit of $\mathbf{x}$. When $\pi = \frac{1}{2}$, we have*
$$\mathbb{E}_{\tilde{\mathbf{x}} \sim p_\pi}[q(\tilde{\mathbf{x}})] = \mathbb{E}_{\mathbf{x} \sim p_{data}}[q(\mathbf{x}_1, \ldots, \mathbf{x}_{k-1})], \qquad (11)$$
*where $q(\mathbf{x}_1, \ldots, \mathbf{x}_{k-1}) := \int_{\mathbf{x}_k} q(\mathbf{x})dx$ is the marginal of $q(\mathbf{x})$ (after marginalizing out $\mathbf{x}_k$).*

*Proof.* To see this, we first note that any $p(\mathbf{x})$ can always be decomposed as $p(\mathbf{x}_1, \mathbf{x}_2, \ldots, \mathbf{x}_k)$. Thus, letting $p_\pi = \mathbb{E}_{x \sim p}[q_\pi(\tilde{\mathbf{x}}|\mathbf{x})]$ (as defined in 5),
$$\mathbb{E}_{\tilde{\mathbf{x}} \sim p_\pi}[q(\tilde{\mathbf{x}})] = \mathbb{E}_{\tilde{\mathbf{x}} \sim q_\pi(\tilde{\mathbf{x}}|\mathbf{x})}\mathbb{E}_{\mathbf{x} \sim p_{data}}[q(\tilde{\mathbf{x}})] \qquad (12)$$
$$= \mathbb{E}_{\mathbf{x} \sim p_{data}}\left[\sum_{i=1}^{2} q(\mathbf{x}_1, \mathbf{x}_2, \ldots, \mathbf{x}_{k-1}, \tilde{\mathbf{x}}_k = i)\right] \qquad (13)$$
$$= \mathbb{E}_{\mathbf{x} \sim p_{data}}[q(\mathbf{x}_1, \ldots, \mathbf{x}_{k-1})], \qquad (14)$$

as desired.

$\qquad\square$

### A.2 ADDITIONAL EXPERIMENTAL DETAILS

Our proposed NCPN architecture consists of the PixelCNN++ backbone Salimans et al. (2017) with axial attention layers Ho et al. (2019b) after each residual block. We retain the hyperparameters of PixelCNN++, changing only the dropout on the CIFAR-10 dataset (from 0.5 to 0.25), which we reduced due to the regularization properties of NCML. For the axial attention layers, we use 8 heads and skip connection rescaling as in Song et al. (2020b). Finally, we add noise conditioning to each residual block via a Gaussian Fourier Projection layer, much like Ho et al. (2020); Song et al. (2020b).

For our NCML-trained models, the diffusion times of the VE, VP, and sub-VP SDEs were chosen to be $T = 0.5$, $T = 0.1$, and $T = 0.025$, respectively. The values are selected such that the standard deviation of the per-pixel differences between samples in $p_{data}$ and their noised counterparts in $p_T$ was $\approx 10$ bits. We suspect that further improvements can be made to the empirical results if these numbers were chosen more judiciously.

| Model | NLL $\pi = 0$* | NLL $\pi = 0.5$ | NLL $\pi = 1$ |
|---|---|---|---|
| NCPN (ML) | 2.25 | 3.72 | 4.35 |
| NCPN (NCML VE) | 2.22 | 3.63 | 4.21 |
| NPCN (NCML sub-VP) | 2.31 | 3.44 | 3.98 |
| NCPN (NCML VP) | 2.48 | 3.14 | 3.67 |

Table 3: Results on CelebA 64x64. Negative log-likelihood (NLL) is in bits per dimension. Lower is better. *NLL with $\pi = 0$ is equivalent to NLL of the original data.

| Data/Model | MLE | NCML-VE | NCML-subVP | NCML-VP |
|---|---|---|---|---|
| SVHN ↑ | 0.35 | 0.43 | 0.65 | **0.74** |
| Const Uniform ↑ | 0.1 | 0.56 | 0.59 | **0.68** |
| Uniform ↑ | **1.0** | **1.0** | **1.0** | **1.0** |
| Average ↑ | 0.48 | 0.66 | 0.73 | **0.80** |
| CIFAR-10 BPD ($\pi = 1$) ↓ | 3.99 | 3.94 | 3.93 | **3.89** |

Table 4: AUROC scores of our NCPN models trained on CIFAR-10 on the OOD detection task, with either MLE or different noise schedules.

All NCPN models were trained on RTX 2080 Ti GPUs for 500,000 iterations. This is approximately 1.5 weeks of training. We use the same NCPN architecture and hyperparameters across all datasets (except for dropout, which is set to 0.25 on CIFAR-10 and 0.00 on ImageNet 64x64 and CelebA 64x64). All NCPN models have 73M parameters.

### A.2.1 Density Estimation and Generative Modeling Experiments

For experiments on CIFAR-10 and ImageNet 64x64, we compare against Kingma et al. (2021); Song et al. (2021); Child (2020); Ho et al. (2019a); Grcić et al. (2021); Salimans et al. (2017); Chen et al. (2018); Child et al. (2019). Some results could not be included due to the irreproducibility of the techniques. There is limited existing work on likelihood-based modeling on CelebA 64x64, so we do not provide comparisons, however the performance of our model is summarized in Table 3.

### A.2.2 Out-of-distribution Detection Experiments

We directly use the CIFAR-10 models trained in A.2.1 and thus retain all hyperparameters from the previous experiment. All NCPN are evaluated with the time condition $t = 0$. Judicious choice of the statistic is important for the performance of the model. For example, Du & Mordatch (2019) use the unnormalized energy function $\log(Z \cdot p(x))$ where $Z = \int \exp f(x)$ is the partition function, and Meng et al. (2020) use $\sum_{i=1}^{d} [\nabla \log p(x)]_i$, where $p(x)_i$ denotes the $i$th coordinate of the score. We use $||\nabla p(x)||_2 = |p(x)| * ||\nabla \log p(x)||_2$. Additionally, we show that performance on the OOD detection task is closely associated with robustness to minimally perturbed data (Table 4), in the sense that improved robustness to $p_\pi$ is associated with improved performance on OOD detection. One possible explanation for this correlation is that robustness to visually inconsequential high frequency content forces the model to rely on more intrinsic features of the image to assign probabilities, which allows the model to better classify OOD images.

### A.3 Additional Figures

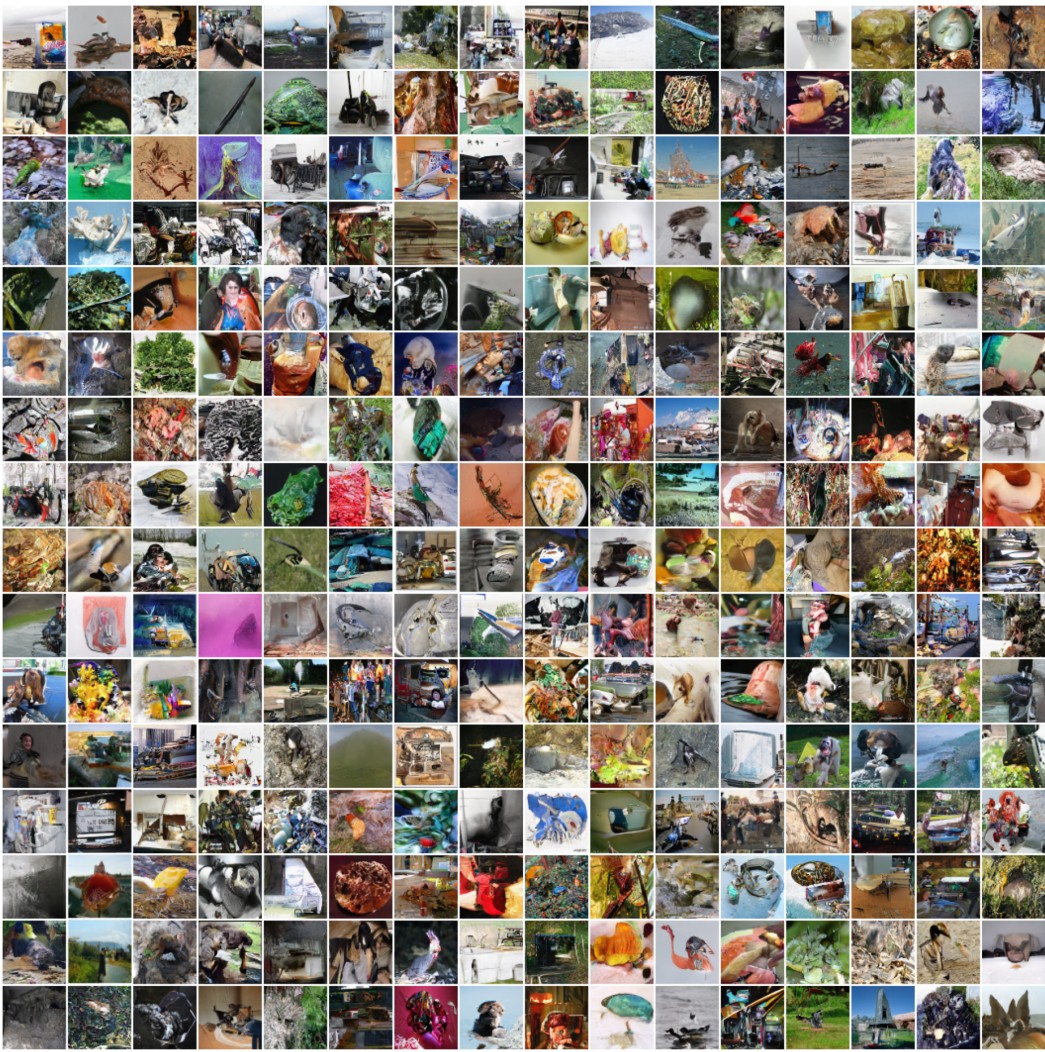

Figure 6: Samples from NCPN trained on ImageNet 64x64, with $p_t$ as a variance preserving (VP) diffusion process.

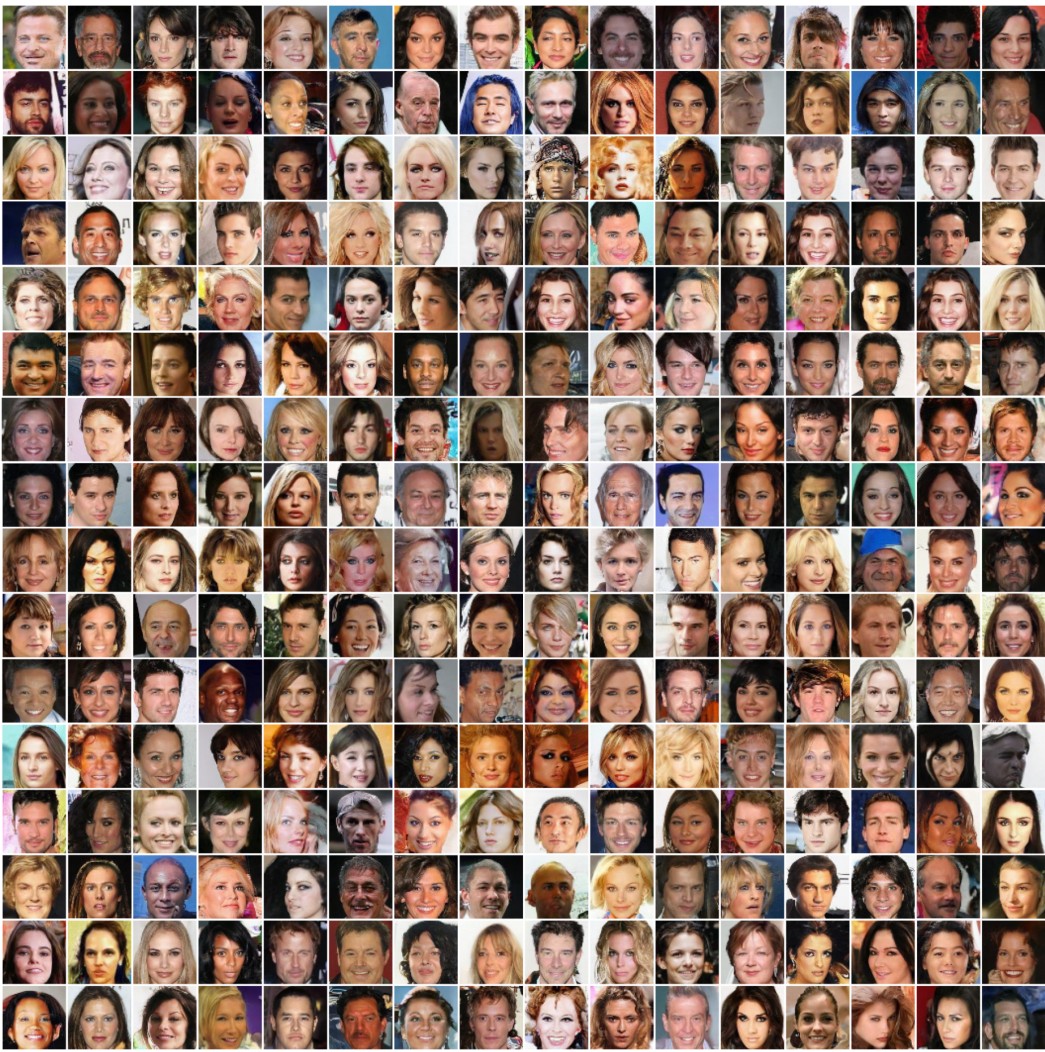

Figure 7: Samples from NCPN trained on CelebA 64x64, with $p_t$ as a variance preserving (VP) diffusion process.

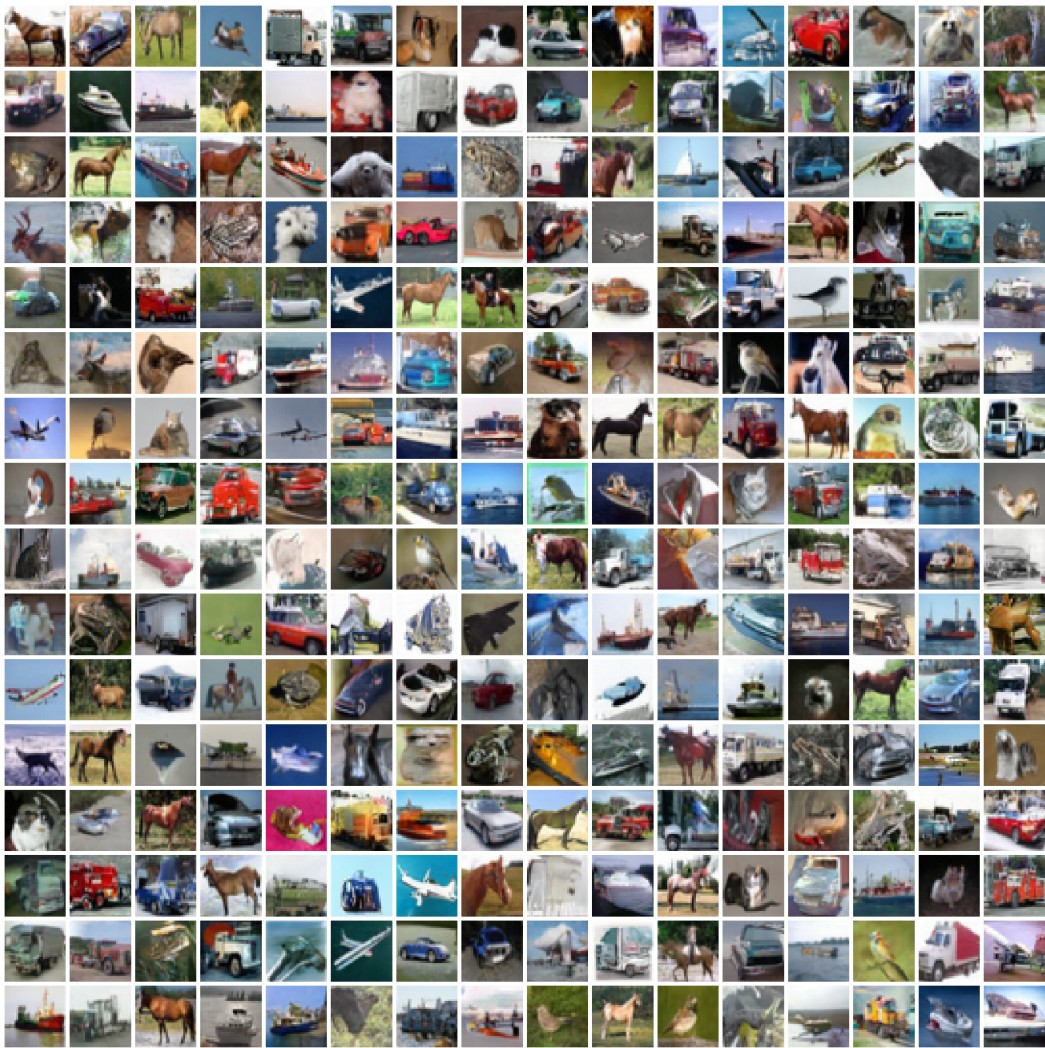

Figure 8: Samples from NCPN trained on CIFAR-10, with $p_t$ as a variance preserving (VP) diffusion process.

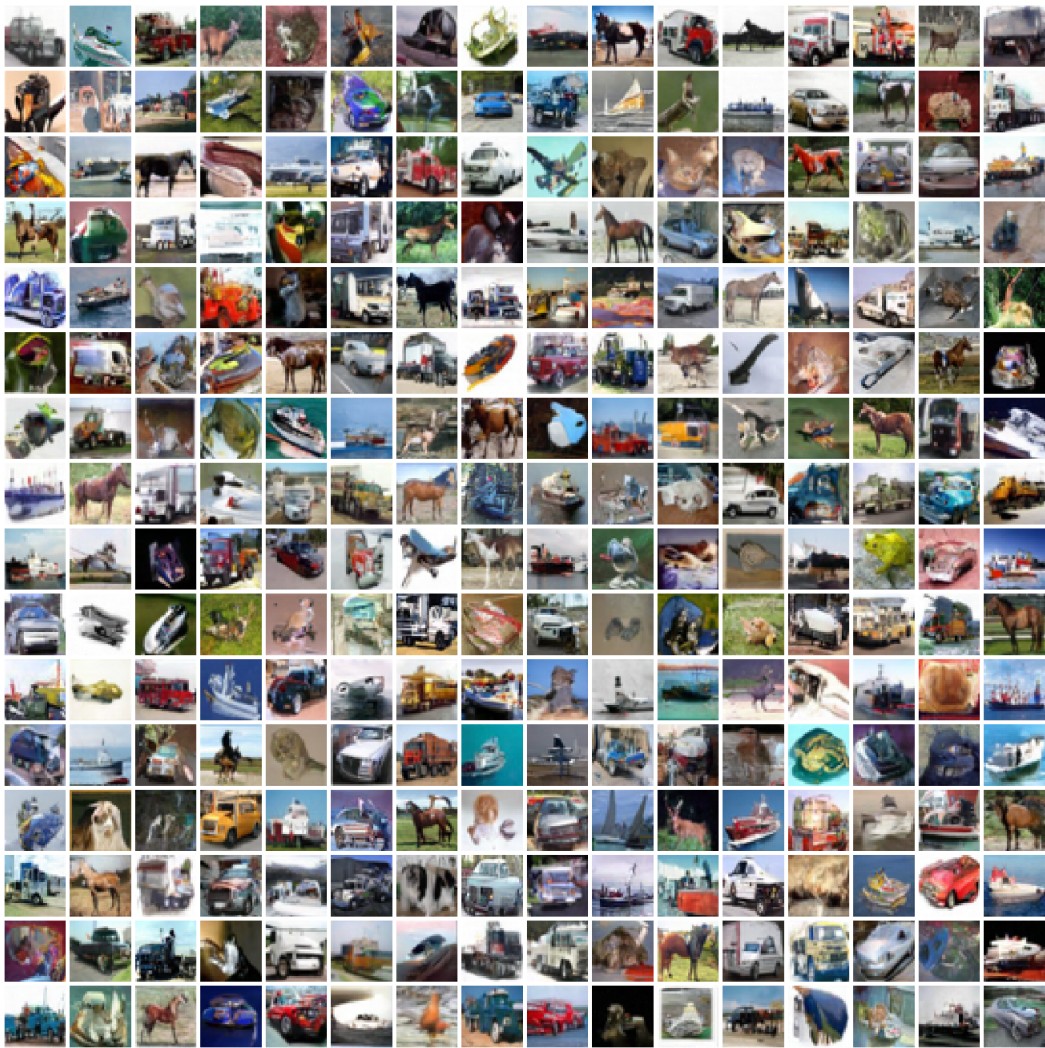

Figure 9: Samples from NCPN trained on CIFAR-10, with $p_t$ as a sub-variance preserving (sub-VP) diffusion process.

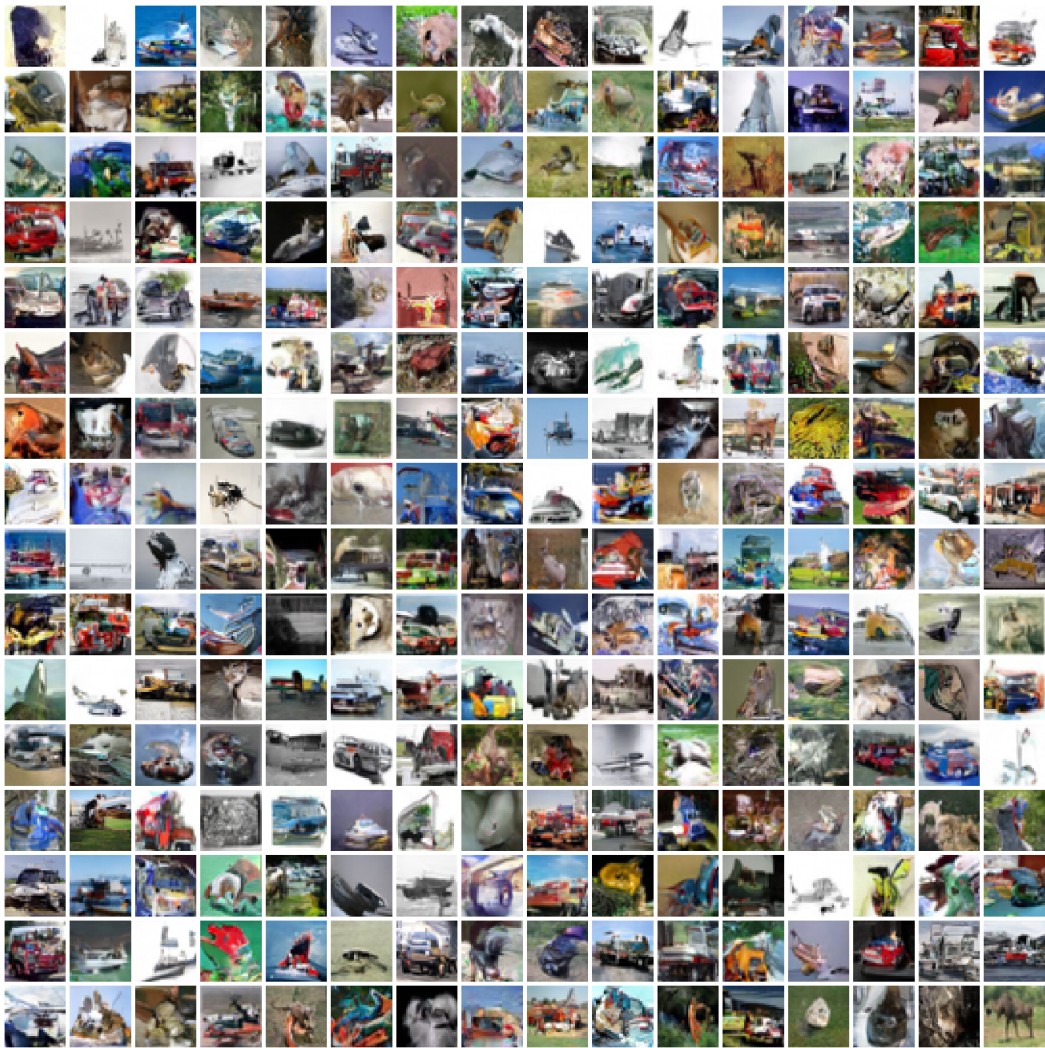

Figure 10: Samples from NCPN trained on CIFAR-10, with $p_t$ as a variance exploding (VE) diffusion process.

