# OpenReview forum: "Autoregressive Generative Modeling with Noise Conditional Maximum Likelihood Estimation"
_ICLR.cc/2023/Conference — Submitted to ICLR 2023_

### Official Review · Reviewer_YYAi · 2022-10-23

**Confidence:** 3
**Correctness:** 4
**Technical Novelty And Significance:** 2
**Empirical Novelty And Significance:** 4
**Recommendation:** 5

**Clarity, Quality, Novelty And Reproducibility:**

The paper is very clearly structured and written.
I believe the idea is completely original, deep and interesting.
Unfortunately, I find that crucial details about the network architecture and training are missing.


**Strength And Weaknesses:**

Strengths:

* The paper is well written. The problem is excellently introduced and motivated.
* The idea is interesting and original as far as I know.

Weaknesses:

* As far as I understand, the argument about autoregressive models being overly sensitive to small changes is based on the assumption, that they use discrete distributions (e.g. for 8bit images) over pixel values, right? I wonder if this would hold for models that use parametric continuous distributions for each pixel.
* I am missing some details in the paper on how exactly the PixelCNN++ architecture was modified.
* How exactly is the model conditioned on the noise level? I think this could make quite a difference in how much more robust the model will become by forcing it to also describe the noisy data.


Minor points:

* The authors write: "p_π describes the distribution of points in pdata that have had their least
significant bit incremented or decremented with probability π"
This formulation could be more clear. What happens when a 0 valued bit is decremented?
Is this simply about adding or subtracting one form the pixel value?


**Summary Of The Paper:**

The paper presents a novel way to train autoregressive models and to sample from them.
The authors argue that the standard maximum likelihood approach is overly sensitive to tiny perturbations, assigning low likelihoods to images that have been modified in a way that is Imperceivable to humans.
Additionally, they argue sampling results from autoregressive models are inferior due to these sensitivities.

The proposed method works by training a model to not only the distribution of the training data, but also of noisy versions of the data.
The model is conditioned on the noise level.
The authors find that their model increases sample quality and is more rout to small perturbations.


**Summary Of The Review:**

All-in-all, this is an interesting paper with some smaller flaws, that could be fixed in the final version.
Most importantly, the details on training and architecture have to be provided.
Finally, I have to admit that I am not an expert on score matching and that somebody with more knowledge of the topic might have a more informed opinion.

I changed my rating regarding the novelty in light of the discussion with other reviewers and the AC.

---

> ### Author Response · Authors · 2022-11-09
> **Response to Reviewer YYAi**
>
> We would like to thank you for your positive review and constructive feedback. We have incorporated your suggestions into our uploaded revision.
>
> **Q: Do we assume images are discrete? Can we apply this to continuous distributions on images?**
>
> **A:** This is a great question. To spoil the conclusion, we already train on continuous distributions. Though the original PixelCNN [1] paper considered learning a discrete softmax distribution over the 8-bit pixel values, work as early as PixelCNN++ [2] have converted to continuous distributions. In fact, our work requires a continuous model over the pixels, otherwise the likelihood is not differentiable, and the score function is undefined. However, all models are essentially trained with the assumption that the pixels are still discrete. (For example, in a monochrome image if pixel 15 has intensity 127, we maximize the likelihood of the model over the interval [126.5, 127.5], thus respecting its discrete nature.) To truly assume continuous distributions, we would need to dequantize the pixels, or obtain images of much higher fidelity, otherwise continuous models will assign infinitely high likelihoods to the discrete integer values of each pixel (see Section 7.3.2 in [3]). However, we suspect that training on truly continuous distributions will not solve the unstable sampling and noise robustness issue, as it is less the discrete nature of the data, and more the spurious high frequency content of the images that is problematic.
>
> [1] Pixel Recurrent Neural Networks. https://arxiv.org/abs/1601.06759
>
> [2] PixelCNN++: Improving the PixelCNN with Discretized Logistic Mixture Likelihood and Other Modifications. https://arxiv.org/abs/1701.05517
>
> [3] Neural Autoregressive Distribution Estimation. https://arxiv.org/abs/1605.02226
>
> **Q: Details on PixelCNN++ architecture modification, and the noise conditioning module.**
>
> **A:** We agree that we did not sufficiently cover the architectural details on the NCPN in the original submission. We have now added this information to the appendix, and also summarized it below.  Specifically, we made two modifications to PixelCNN++:
> * Inspired by the attention layers in NCSN and DDPM models [1, 2], we added axial attention layers [3] after each residual block. This sparse attention mechanism is closely related to that used in Sparse Transformers [4] and is similarly much lighter in computational cost than full attention layers, but does not require any special hardware or CUDA kernels.
> * We add noise conditioning to each residual block via the Gaussian Fourier Projection layer proposed in [1, 2].
>
> Finally, we retain all hyperparameters of PixelCNN++, changing only the dropout on the CIFAR-10 dataset (from 0.5 to 0.25), which we reduced due to the regularization properties of NCML.
>
> [1] Denoising Diffusion Probabilistic Models. https://arxiv.org/abs/2006.11239
>
> [2] Score-Based Generative Modeling through Stochastic Differential Equations. https://arxiv.org/abs/2011.13456
>
> [3] Axial Attention in Multidimensional Transformers. https://arxiv.org/abs/1912.12180
>
> [4] Generating Long Sequences with Sparse Transformers. https://arxiv.org/abs/1904.10509
>
> **Q: Details on $p_\pi$. Does the perturbation simply involve adding or subtracting one from the pixel value?**
>
> **A:** ~~Yes, you are correct. We do see how using "increment" and "decrement" terminology in conjunction with bits can be misleading. We are not referring to bit arithmetic, but rather simple integer arithmetic. We have revised the terminology in Section 3.1 to clarify this.~~
>
> EDIT: Following the suggestion of Reviewer 4m9N, we have actually changed the definition of $p_\pi$ so that it is indeed a bit flip operation on the least significant bit. It turns out that this change is mainly cosmetic, as the experimental results remain almost entirely unchanged (we have updated them nonetheless). But it does allow for us to state Lemma 3.2. We have revamped Section 3 by clarifying the definitions in 3.1 and formalizing the arguments in 3.2, so please take a look if you are interested.

---

> > ### Comment · Reviewer_YYAi · 2022-11-18
> > **Thanks!**
> >
> > I believe all my questions have been addressed.
> > I will not change my rating.

---

### Official Review · Reviewer_4nNY · 2022-10-23

**Confidence:** 2
**Correctness:** 3
**Technical Novelty And Significance:** 3
**Empirical Novelty And Significance:** 3
**Recommendation:** 8

**Clarity, Quality, Novelty And Reproducibility:**

I am only familiar with the major works in the diffusion and autoregressive image modelling literature. I am thus unable to confidently evaluate most of these criteria, except for saying that the writing seems clear. With this pinch of salt, my questions and comments are below:

* Given how brittle you find the current image likelihood image models to be, I find the motivation for seeking a better class of likelihood models towards the bottom of page 3—OOD and adversarial defence—somewhat unconvincing. (Especially given that the cited papers were presumably using models that presumably suffer from the brittleness you identify.) Given the remaining gap to score-based models (& their ability to evaluate likelihood), can you provide an alternative motivation (or refute the above) please? Do you expect it to be harder to learn the log density itself, or its gradient (as in score-based models)?

* Relatedly, arguments such as https://arxiv.org/abs/2012.03808 showing that density modelling is not sufficient for tasks like OOD. A way to address this is to be more careful about choice of the space in which likelihood modelling is done. Could the brittleness you identified be addressed by modelling likelihood in some continuous or discrete (VQ) latent space, rather than the pixel-space?

* Do you think similar brittleness issues exist for autoregressive **text** models?

* In the second difference from diffusion models you describe on p. 6, you say that your model does not have to ensure that $p_T$ is an appropriate stationary distribution (as is the case for diffusion models). Can you please explain whether your sampling scheme based on solving a reverse SDE would still be valid if you chose a different $p_T$ for your model?

NIT: In equation (2) and elsewhere, shouldn't it be gradients of the **log** density (see, e.g., the cited paper by hyvarinen)?


**Strength And Weaknesses:**

### strengths
* interesting observations regarding brittleness of modern pixel-space autoregressive image models
* proposed model does seem to provide improvements over other models in the same class

### weaknesses
* the proposed model still lacks significantly behind SOTA diffusion models
* somewhat weak motivation for why we would want to have an explicit likelihood model

**Summary Of The Paper:**

This paper investigates a likelihood model for a family of increasingly noisy distributions, with the true data distribution at one end, and noise at the other. This construction is conceptually similar to the popular class of diffusion models, except the authors decided to learn a log density model of the data (and its noisy versions), instead of just learning its derivative (cf. the score-based interpretation of diffusion models). The authors motivate learning density for the whole family of noisy distributions—instead of just for the data—by showing that modern pixel space autoregressive image models are rather brittle to bit flip perturbations that are barely perceptible by human eye. Due to equivalence to diffusion processes, the authors propose a sampling process based on integrating the SDE in reverse as known from the diffusion literature. Their empirical results of their models improve on other explicit log likelihood models in terms test log likelihood, albeit still lack significantly behind the FID scores of SOTA diffusion models.

**Summary Of The Review:**

While the empirical results are somewhat lacking behind score-based models, they, I found the paper an insightful (especially the part on brittleness of modern autoregressive likelihood models) and enjoyable read.

---

> ### Author Response · Authors · 2022-11-09
> **Response to Reviewer 4nNY (2/2)**
>
> **Q: Brittleness in likelihood-based text models.**
>
> **A:** Indeed, sampling instability in text-based models [1, 2, 3] is quite well documented. However, efforts to mitigate or remedy these instabilities have been largely heuristic (see cited), and break the MLE framework. While this is encouraging for our work, our approach cannot be directly applied to text, as the space is discrete, so the log probability is not differentiable and the score is not well defined. We do believe it would be a very exciting direction for further work.
>
> [1] Scheduled Sampling for Sequence Prediction with Recurrent Neural Networks. https://arxiv.org/abs/1506.03099
>
> [2] How (not) to Train your Generative Model: Scheduled Sampling, Likelihood, Adversary? https://arxiv.org/abs/1511.05101
>
> [3] Professor Forcing: A New Algorithm for Training Recurrent Networks. https://arxiv.org/abs/1610.09038
>
> **Q: Will the sampling scheme still be valid if you picked a different $p_T$ for your model?**
>
> **A:** Yes. Note that, keeping the SDE fixed, $p_T$ is simply a function of $T$. Therefore, letting $T'$ be the new stopping time, as long as the model has been trained on the new interval $[0, T']$ (i.e. $T' < T$), the reverse SDE can be solved by simply changing the initial condition. If $T' > T$, then the model must additionally be trained over the new interval $[0, T']$. If we also change the SDE, then the new solution further requires a time remapping function $f:\mathbb{R} \rightarrow \mathbb{R}$ where the new and old noise levels correspond $\sigma'(f(t')) = \sigma(t)$ so that the model can be appropriately reparameterized w.r.t. the time parameter.
>
> **Q: NIT: In equation (2) and elsewhere, shouldn't it be gradients of the log density?**
>
> **A:** Indeed, you are absolutely correct. Thank you for this catch. We have updated the text accordingly.

---

> ### Author Response · Authors · 2022-11-09
> **Response to Reviewer 4nNY (1/2)**
>
> Thank you for your insightful questions and encouraging feedback! We have incorporated your suggestions into our uploaded revision. We additionally hope that we have addressed your concerns below.
>
> **Q: The proposed model still lacks significantly behind SOTA diffusion models.**
>
> **A:** This is indeed the case in terms of sample quality (i.e., FID). However, NCML models show significant improvements over diffusion models on the ImageNet 64x64 dataset in terms of density estimation, and boast much faster likelihood evaluations (by at least 30x and up to 1000x).
>
> **Q: The motivation for working on a better class of likelihood models is unconvincing, especially given their brittleness and the remaining gap to score-based models. Moreover, [1] suggests that likelihoods are not sufficient for tasks like OOD detection.**
>
> **A:** Indeed, likelihoods can be brittle, and models trained by maximum likelihood estimation are not state of the art in terms of sample quality. However, the gap is much smaller (or non-existent) in terms of likelihoods, and ---to us--- the fact that these models perform reasonably in spite of the shortcomings invites further investigation.
>
> To demonstrate how our noise conditional maximum likelihood estimation framework can improve performance on downstream tasks, we consider an OOD benchmark initially conceived by [2, 3] and extended by [4, 5], where probabilistic models trained on CIFAR-10 are tasked to distinguish between in-distribution points (i.e., CIFAR-10 images) and out-of-distribution points (SVHN images, constant value images, and uniformly distributed images). Noise conditional training greatly improves the robustness of autoregressive likelihood models. We believe that similar improvements can be explored in other settings, though this endeavor lies outside the scope of this paper. We have added this particular experiment to the Appendix in the text, along with further experimental details.
>
> | OOD Data / Model 	| PixelCNN++	| GLOW	| EBM	| AR-CSM	| NCPN (Ours)	|
> |---|---|---|---|---|---|
> | SVHN				| 0.32		| 0.24	| 0.63	| 0.68		| **0.74**			|
> | Const Uniform		| 0.0			| 0.0		| 0.30	| 0.57		| **0.68**			|
> | Uniform				| **1.0**			| **1.0**		| **1.0**		| 0.95		| **1.0**			|
> | Average			| 0.44		| 0.41	| 0.64	| 0.73		| **0.80**			|
>
> Finally, our sanity test experiments suggest that much of earlier gains in likelihood modeling have focused on modeling the least significant bit (LSB), which we find to contain information of little visual significance (Figure 2). On the other hand, our models perform much better on the sanity test experiment. We have added Lemma 3.2, which more rigorously describes the relationship between our test and the likelihood model on the non-LSB information content in the image.
>
> [1] Perfect density models cannot guarantee anomaly detection. https://arxiv.org/abs/2012.03808
>
> [2] Do Deep Generative Models Know What They Don't Know? https://arxiv.org/abs/1810.09136
>
> [3] Deep Anomaly Detection with Outlier Exposure. https://arxiv.org/abs/1812.04606
>
> [4] Autoregressive Score Matching. https://arxiv.org/abs/2010.12810
>
> [5] Implicit Generation and Generalization in Energy-Based Models. https://arxiv.org/abs/1903.08689
>
> **Q: Likelihood modeling in a continuous or discrete (VQ) latent space, rather pixel space.**
>
> **A:** To our knowledge, it is still an open question as to how one can compute the pushforward measure of a non-invertible transformation, while retaining computational tractability and expressiveness of the transformation. Latent spaces are usually much lower dimension than the data. Therefore, it would be difficult to simultaneously build a generative model that can also evaluate likelihoods on a more informative latent space. However, this is indeed an interesting direction worth considering.

---

### Official Review · Reviewer_Vbpt · 2022-10-24

**Confidence:** 5
**Correctness:** 2
**Technical Novelty And Significance:** 2
**Empirical Novelty And Significance:** 2
**Recommendation:** 3

**Clarity, Quality, Novelty And Reproducibility:**

**Clarity**
I may be misunderstanding Section 3, but isn't corrupting the least significant bit of a pixel value just changing the high frequency content in an image? If this is the case, I'm not sure of the relevance of this approach. It is known that most of the information content (i.e. entropy) of an image is contained in the high-frequency detail (e.g. see variational diffusion models and their results which show most of the BPD contribution happens near data). This is potentially a reason why the epsilon-parameterization of diffusion models has become so popular, in that it (exponentially) downweights the loss near data and therefore cares much less about high-frequency detail which is much less relevant for human perception (and perhaps as you show, can confuse models). I feel the motivation of the LSB perturbation scheme, and why it captures some meaningful invariance we should practically care about (as opposed to showcasing a failure mode of likelihood measurement for high-dimensional image data), are both unclear.

**Originality**
A related result is presented in https://openreview.net/forum?id=rJA5Pz7lHKb, where the authors fit noisy data using an AR model, and then fit clean data conditioned on noisy data. This leads to considerable improvements in sample quality and likelihood evaluation. Though not quite the same as the setup here, I feel this paper deserves a mention, but it is not discussed or cited.

**Strength And Weaknesses:**

# Strengths
The paper demonstrates that training an autoregressive model as the marginal t = 0 of a noise process improves sample generation at no extra cost.

# Weaknesses

**Is likelihood really important?**
It is arguable the degree to which we should care about the likelihood (and ability to evaluate the likelihood) in and of itself, despite the paper's arguments. For example:
- 2.1: 'therefore do not have the same asymptotic guarantees as MLE (e.g., efficiency, functional equivariance, sufficiency).' Score-matching also has asymptotic guarantees. Moreover, how much do we really believe in the guarantees of MLE in practice? For example, the parameters of any neural-network-based model will not be identifiable, meaning most of the guarantees of MLE don't apply.
- 2.1: 'Moreover, likelihoods have no closed-form expression, thus requiring approximate ODE/SDE solvers and hundreds to thousands of function evaluations to evaluate.'
If we don't need likelihoods to train, why does it matter that they're expensive to evaluate? There is no key functionality presented in the paper that relies on efficient likelihood estimation.

Usually likelihood evaluation, be it exact, approximate, or through a bound, is used as a training objective for generative models. If we don't need it to train, why do we care about it? I feel this isn't well-motivated.

**Questionable claims about generative models**
The paper states more than once that diffusion models cannot be trained by maximum likelihood.
Intro: 'diffusion models are poor likelihood models, as they cannot be trained via maximum likelihood.'
2.1: 'However, they cannot be trained by maximum likelihood'
This is incorrect. Diffusion models formulated either as latent-variable models or as score-based models can both be fit by maximum likelihood. There is a footnote at the end of page one to say 'at best they can optimize a lower bound of the likelihood', but the main statements claiming (i) diffusion models are poor likelihood models and (ii) diffusion models cannot be trained by maximum likelihood are incorrect. Not only can score-based models be trained by maximum likelihood, but they are also effectively continuous-time normalizing flows, but can be trained more efficiently.

2.1: 'Likelihood models draw samples x ∼ pθ one of two ways.'
Stating that likelihood-based models draw samples either as discrete-time flows or as AR models is reductive. This fails to include continuous-time flows or latent-variable models, and diffusion models can be formulated as either.

**Experimental results**
It is unclear the degree to which the model's robustness to the LSB perturbation scheme is practically useful. Moreover, the proposed score-based sampling scheme is strictly more expensive than standard AR sampling, since sampling requires D passes of the model for the initial noise sample, then taking gradients through a single evaluation of the model for each subsequent score evaluation. As mentioned before, I'm not sure of the usefulness of the likelihood measurements under LSB-perturbation for various models in Table 1, and the single FID value reported for CIFAR-10 along with qualitative samples is unconvincing.

**Nits**
- Intro: 'AR models boast state-of-the-art performance in many domains, including images... and audio'. This is certainly not true for images anymore, and is it true for audio?

- The paper repeatedly states the likelihood is a function of the data when it is a function of the parameters of the model.
Abstract: 'Likelihood of the data under the model'
2.1: 'the likelihood of each sample'
3.2: 'the model’s predicted likelihood of downstream tokens'

- 'The goal of likelihood-based generative modeling is to approximate pdata via a parametric model pθ, where samples x ∼ pθ can be easily obtained.' This is the goal of generative modeling, not likelihood-based generative modeling.

**Summary Of The Paper:**

This paper proposes to train noise-conditional autoregressive models across a continuum of noise levels, analogously to diffusion models. The authors argue that such models improve density estimation performance under a least-significant-bit perturbation scheme, and also improve sample generation.

**Summary Of The Review:**

I feel the characterization of various generative models is lacking throughout the paper, and the contribution is poorly motivated and limited in experimental evaluation. Overall, I don't think the paper does enough to recommend acceptance.

---

> ### Author Response · Authors · 2022-11-09
> **Response to Reviewer Vbpt (5/5)**
>
> **Q: 'The goal of likelihood-based generative modeling is to approximate pdata via a parametric model pθ, where samples x ∼ pθ can be easily obtained.' This is the goal of generative modeling, not likelihood-based generative modeling.**
>
> **A:** You are correct, as all generative models at least implicitly approximate $p_{data}$. We have edited this sentence to read: The goal of likelihood-based generative modeling is to approximate pdata via a parametric model $p_\theta$, **where $p_\theta$ can be explicitly evaluated**, and samples $\mathbf{x} \sim p_\theta$ can be easily obtained.
>
>
> **Q: It is known that most of the information content (i.e. entropy) of an image is contained in the high-frequency detail (e.g. see variational diffusion models and their results which show most of the BPD contribution happens near data). I feel the motivation of the LSB perturbation scheme, and why it captures some meaningful invariance we should practically care about (as opposed to showcasing a failure mode of likelihood measurement for high-dimensional image data), are both unclear.**
>
> **A:** Indeed, measures of statistical entropy prioritize information content in the least significant bit (LSB).  However, even though this suggests that models are more inclined to model the LSB (and thus waste model capacity on visually insignificant features), it does not mean we must submit themselves to this undesirable outcome. There are ways to reduce this modeling bias. This is precisely our goal with NCML, and what our noise robustness experiments highlight. Following reviewer 4m9N's suggestion, we formalize the intuitions in Section 3, namely the connection between noise-robustness under our test and modeling performance on the non-LSB information of the image. Lemma 3.2 suggests that better performance on the noise-robustness tests means better modeling of non-LSB information, and thus the visually salient content in the image. **In short, we motivate the LSB perturbation scheme as a likelihood-based measure related to sample quality, and show that our models perform better on this measure.**
>
> **Q: Related result: [1].**
>
> **A:** Thank you for bringing this to our attention. Reviewer 4m9N also mentioned this text. We have added this work to our paper. Some differences:
> * The decomposition proposed by [1] introduces latent variables to the probabilistic model, which means that it can no longer evaluate the likelihood of the generative model. As a result, the authors resort to ELBO optimization for training. Our model retains likelihood evaluation and maximum likelihood estimation capabilities.
>  * [1] only considers a single denoising step for generation, which must be defined at training time. Our model can adaptively sample and denoise from multiple noise levels, depending on available computational resources.
> * The denoising step in [1] is autoregressive and requires 32x32=1024 evaluations of the underlying network on CIFAR-10. Conversely, each denoising step in our model requires only a single gradient evaluation of the network. Therefore our denoising process is ~50x faster than [1].
>  * We achieve much better sample quality and density estimation performance in terms of FID and BPD.
>
> [1] Improved Autoregressive Modeling with Distribution Smoothing. https://arxiv.org/abs/2103.15089

---

> ### Author Response · Authors · 2022-11-09
> **Response to Reviewer Vbpt (4/5)**
>
> **Q: I'm not sure of the usefulness of the likelihood measurements under LSB-perturbation for various models in Table 1, and the single FID value reported for CIFAR-10 along with qualitative samples is unconvincing.**
>
> **A:** Our experiments and theory suggest that robustness to LSB perturbations is correlated with improved sample quality in autoregressive models (Table 1) and better modeling of visually significant features of the image (Section 3.2). As additional evidence, we extend the aforementioned OOD experiment to further investigate the correlation between LSB perturbation robustness and practical robustness measures, such as OOD detection robustness. We compare our NCPN model trained via maximum likelihood estimation (MLE) and NCML trained with a variance exploding (VE), variance preserving (VP), and sub-VP stochastic process. From this experiment, we see an expected correlation between robustness to the least significant bit and better performance. One explanation for this correlation is that robustness to visually inconsequential high frequency content forces the model to rely on more intrinsic features of the image to assign probabilities, which allows the model to better classify OOD images (Lemma 3.2).
>
> | Task / Model 		| MLE		| NCML (VE)	| NCML (sub-VP)	| NCML (VP)	|
> |---|---|---|---|---|
> | OOD: SVHN			| 0.35		| 0.43		| 0.65			| **0.74**		|
> | OOD: Const Uniform	| 0.1			| 0.56		| 0.58			| **0.68**		|
> | OOD: Uniform		| **1.0**	| **1.0**	| **1.0**		| **1.0**			|
> | OOD: Average		| 0.48		| 0.66		| 0.74			| **0.80**		|
> | BPD: CIFAR ($\pi = 1$)	| 3.99		| 3.94		| 3.93			| **3.89**		|
>
> **Q: Intro: 'AR models boast state-of-the-art performance in many domains, including images... and audio'. This is certainly not true for images anymore, and is it true for audio?**
>
> **A:** We see how this can be contentious, as MLE models do achieve state-of-the-art performance in some areas (e.g., we achieve state-of-the-art average log likelihood on ImageNet64x64) while being inferior in others (diffusion models generally beat AR models on sample quality). Thus we've changed "state-of-the-art" to "competitive".
>
> **Q: The paper repeatedly states the likelihood is a function of the data when it is a function of the parameters of the model. Abstract: 'Likelihood of the data under the model' 2.1: 'the likelihood of each sample' 3.2: 'the model’s predicted likelihood of downstream tokens'.**
>
> **A:** We were not aware of this convention on describing the likelihood. By our understanding, the likelihood assigns a probability score to (data, parameter) pairs, which are proportional to how well the data is described by the model parameters, and is thus a function of both. It does appear that the likelihood is sometimes written as $L(\theta|X)$ (i.e. emphasizing that it is a function of the parameters given the data) [1]. However, many works in machine learning appear to write the likelihood as $p(x|\theta)$ ([2, 3, 4, 5, 6, 7] (i.e., data given the parameters), perhaps because the likelihood is often decomposed as a function of $x$ for computational tractability (e.g. change-of-variables, conditional product rule, probability flow ODE). However, we do agree that this can be confusing. To reduce confusion for adherents to either convention, we have removed statements in the text of the form "likelihood of [...]".
>
> [1] Statistical Inference. https://mybiostats.files.wordpress.com/2015/03/casella-berger.pdf
>
> [2] Latent Dirichlet Allocation. https://www.jmlr.org/papers/volume3/blei03a/blei03a.pdf
>
> [3] Pixel Recurrent Neural Networks. https://arxiv.org/pdf/1601.06759.pdf
>
> [4] PixelCNN++: Improving the PixelCNN with Discretized Logistic Mixture Likelihood and Other Modifications. https://arxiv.org/pdf/1701.05517.pdf
>
> [5] Density Estimation Using RealNVP. https://arxiv.org/pdf/1605.08803.pdf
>
> [6] Score-Based Generative Modeling through Stochastic Differential Equations. https://arxiv.org/abs/2011.13456
>
> [7] Variational Diffusion Models. https://arxiv.org/pdf/2107.00630.pdf

---

> ### Author Response · Authors · 2022-11-09
> **Response to Reviewer Vbpt (3/5)**
>
> **Q: Score-based models are trained by maximum likelihood.**
>
> **A:** We make clear a distinction between optimization processes that increase likelihoods (i.e., ELBO maximization) and maximum likelihood estimation (MLE). MLE is not simply any optimization process that increases the model likelihood. Rather, it is a specific technique for parameter estimation via direct maximization of the likelihood. While ELBO maximization also increases the likelihood via a lower bound, it provides no asymptotic guarantees on the correctness of the resulting model, for general parametric models. Moreover, maxima for ELBO may not even coincide with maxima for the model log likelihood. **In other words, solutions that maximize ELBO are not necessarily solutions of maximum likelihood**. Therefore, diffusion models (which are trained with denoising score matching or ELBO optimization) are theoretically distinct from our model. We do see how this confusion can occur, as this distinction is further blurred by a recent and influential paper [1], which describes ELBO maximization as maximum likelihood.
>
> To help reduce this confusion, we have changed all instances in the paper of "maximum likelihood" and "likelihood maximization models" to "maximum likelihood estimation" and "MLE-based models", so as to more explicitly highlight this distinction. We have also emphasized this distinction in Sections 1 and 2.
>
> [1] Maximum Likelihood Training of Score-Based Diffusion Models. https://arxiv.org/abs/2101.09258
>
> **Q: Score-based models are effectively continuous-time normalizing flows, but can be trained more efficiently.**
>
> **A:** They are normalizing flows. But this does not mean that they are trained via MLE. They are only trained by denoising score matching / ELBO maximization.
>
>
> **Q: The characterization of likelihood models in Section 2.1 is reductive, and does not contain continuous normalizing flows or latent variable models.**
>
>  **A:** We note that continuous normalizing flows (CNFs) still draw samples the same way as discrete flows, through a series of invertible transformations applied to a prior, though the number of transformations is now (theoretically) infinite. However, we do agree that the citation provided in this section is misleading, as we only cite a discrete flow. We now additionally cite [1] to rectify this. Latent variable models, on the other hand, are usually not likelihood models, as they cannot explicitly represent a data likelihood, even though they are often trained via optimizing a lower bound (the ELBO) --- one exception, of course, being diffusion models which we discuss at length.
>
> [1] FFJORD: Free-form Continuous Dynamics for Scalable Reversible Generative Models. https://arxiv.org/abs/1810.01367
>
> **Q: It is unclear the degree to which the model's robustness to the LSB perturbation scheme is practically useful.**
>
> **A:**
> Unstable sequential sampling due to compounding sampling errors is well-documented in autoregressive generative models [1, 2, 3] (also see Figure 1). Thus, the noise-robustness experiments serve to quantify robustness to such errors. The proposed sanity test can be seen as measuring the robustness to errors in the least significant bit (LSB). (See Lemma 3.2.) In 8-bit images, the LSB contains image features that are almost entirely visually inconsequential (Figure 2), and perturbing it constitutes the smallest possible error that can be effected on an image. **Therefore, the proposed sanity test serves two purposes**: First, it provides a necessary condition for stable sampling. Second, it measures modeling performance on the non-LSB (ie more visually important) information in an image.
>
> [1] Improved Autoregressive Modeling with Distribution Smoothing. https://arxiv.org/abs/2103.15089
>
> [2] Scheduled Sampling for Sequence Prediction with Recurrent Neural Networks. https://arxiv.org/abs/1506.03099
>
> [3] Professor Forcing: A New Algorithm for Training Recurrent Networks. https://arxiv.org/abs/1610.09038
>
> **Q: Moreover, the proposed score-based sampling scheme is strictly more expensive than standard AR sampling, since sampling requires D passes of the model for the initial noise sample, then taking gradients through a single evaluation of the model for each subsequent score evaluation.**
>
>  **A:** Indeed, there is an added cost to our refinement scheme. However, the number of steps for refinement can be dynamically adjusted. (For example, it can even be set to 0.) Moreover, it is generally lower than competing works on improved autoregressive modeling, e.g., [1]. For the model that attains an FID of 12.09, we choose 100 refinement steps. Compare with [1], which attains an FID of 29.83 and requires resampling every single pixel in the denoising step, i.e. 32 x 32 = 1024 refinement steps for CIFAR-10. We have a 10x improvement on the number of steps, plus a more than 2x reduction in FID.
>
> [1] Improved Autoregressive Modeling with Distribution Smoothing. https://arxiv.org/abs/2103.15089

---

> ### Author Response · Authors · 2022-11-11
> **Response to Reviewer Vbpt (2/5)**
>
> **Q: There is no key functionality presented in the paper that relies on efficient likelihood estimation.**
>
> **A:** Indeed, our original submission did not include any such functionality, as we focus on the main goal of generative modeling and density estimation.
>
> However, efficient likelihood estimation is essential for many downstream tasks, such as out-of-distribution (OOD) detection, uncertainty estimation, and adversarial defense. To demonstrate this, we have included additional experiments on an Out-of-Distribution (OOD) detection baseline. We use a benchmark initially conceived by [1] and extended by [2, 3, 4], where probabilistic models trained on CIFAR-10 are tasked to distinguish between in-distribution images (i.e., CIFAR-10 images) and out-of-distribution images (SVHN images, constant value images, and uniformly distributed images). Pixel autoregressive likelihood models are notorious for their poor performance in this task [1, 2]. Our noise conditional training greatly improves their robustness, and we obtain SOTA results on this benchmark. We believe that this experiment further improves our empirical results, and have added it to the text, along with further experimental details.
>
> | OOD Data / Model 	| PixelCNN++	| GLOW	| EBM	| AR-CSM	| NCPN (Ours)	|
> |---|---|---|---|---|---|
> | SVHN				| 0.32		| 0.24	| 0.63	| 0.68		| **0.74**			|
> | Const Uniform		| 0.0			| 0.0		| 0.30	| 0.57		| **0.68**			|
> | Uniform				| **1.0**			| **1.0**		| **1.0**		| 0.95		| **1.0**			|
> | Average			| 0.44		| 0.41	| 0.64	| 0.73		| **0.80**			|
>
> [1] Do Deep Generative Models Know What They Don't Know? https://arxiv.org/abs/1810.09136
>
> [2] Deep Anomaly Detection with Outlier Exposure. https://arxiv.org/abs/1812.04606
>
> [3] Autoregressive Score Matching. https://arxiv.org/abs/2010.12810
>
> [4] Implicit Generation and Generalization in Energy-Based Models. https://arxiv.org/abs/1903.08689
>
> **Q: Are likelihoods really important? Usually likelihood evaluation, be it exact, approximate, or through a bound, is used as a training objective for generative models. If we don't need it to train, why do we care about it?**
>
> **A:** Likelihoods are, in fact, used outside of training. A key result in [1] was that it attained competitive likelihoods while producing images of high sample quality, even though the model itself did not train with likelihoods. [2] centrally claims to perform faster sampling in diffusion models, without losing the capability for likelihood evaluations that many faster sampling algorithms sacrifice. In general, many works in generative modeling via variational lower bound maximization [3, 4, 5] report likelihoods, even though likelihoods are not actually used to train (e.g., training is done on the lower bound, and likelihoods must be explicitly calculated at test time). Finally, likelihoods models themselves can be used for important tasks such as OOD detection and adversarial defense [6, 7, 8].
>
> [1] Score-Based Generative Modeling through Stochastic Differential Equations. https://arxiv.org/abs/2011.13456
>
> [2] Subspace Diffusion Generative Models. https://arxiv.org/abs/2205.01490
>
> [3] Auto-Encoding Variational Bayes. https://arxiv.org/abs/1312.6114
>
> [4] Variational Diffusion Models. https://arxiv.org/abs/2107.00630
>
> [5] NVAE: A Deep Hierarchical Variational Autoencoder. https://arxiv.org/abs/2007.03898
>
> [6] Density Ratio Estimation via Infinitesimal Classification. https://arxiv.org/abs/2111.11010
>
> [7] Likelihood Ratios for Out-of-Distribution Detection. https://arxiv.org/abs/1906.02845
>
> [8] PixelDefend: Leveraging Generative Models to Understand and Defend against Adversarial Examples. https://arxiv.org/abs/1710.10766

---

> ### Author Response · Authors · 2022-11-11
> **Response to Reviewer Vbpt (1/5)**
>
> We greatly appreciate your extensive feedback. Respectfully, we find that some of your concerns may be misinformed, and have arisen from confusing terminology in the current literature. Clarifying these misunderstandings, as well as incorporating your other feedback in our paper has holistically improved its clarity and readability.
>
> **Q: Score-matching also has asymptotic guarantees.**
>
> **A:** Indeed, score-matching also has some, but not all, of the asymptotic guarantees of maximum likelihood estimation (namely consistency [1, 2] and asymptotic normality [2]). However, the generative models obtained by denoising / diffusion frameworks (i.e., via the SDE or probability flow ODE that can be generated by the learned scores [3]) **do not have such guarantees**. This is because the final model is obtained by integrating over all learned noise conditional scores according to a user-specified noise schedule. While each conditional score at each noise level possesses the asymptotic guarantees of score matching, the resulting diffusion model does not. Thus the asymptotic guarantees of score matching no longer hold. (To see this, note that the noise schedule used to integrate the reverse diffusion ODE/SDE can be arbitrary, and its choice greatly influences the resulting model. Moreover, this schedule is not prescribed by the score matching framework. Therefore score matching cannot directly produce a diffusion model, and so consistency and asymptotic normality cannot possibly hold.)
>
> We do see how this can be easily misinterpreted, and have added clarification to Sections 1 and 2 to mitigate such confusions.
>
> [1] Sliced Score Matching: A Scalable Approach to Density and Score Estimation. https://arxiv.org/abs/1905.07088
>
> [2] Estimation of Non-Normalized Statistical Models by Score Matching. https://www.jmlr.org/papers/volume6/hyvarinen05a/hyvarinen05a.pdf
>
> [3] Score-Based Generative Modeling through Stochastic Differential Equations. https://arxiv.org/abs/2011.13456
>
>
> **Q: How much do we really believe in the guarantees of MLE in practice? Neural networks are generally non-(point-)identifiable.**
>
> **A:** While neural networks are indeed not point-identifiable, neural networks are still set identifiable. Identifiability analysis has its roots in econometrics so we refer to [1] for a more general treatment of this topic, though it has recently been applied to machine learning in nonlinear ICA models [2], energy based models [3], and more general discriminative models [4]. Indeed, overparameterized models (such as neural networks) are usually not identifiable (or what is now commonly referred to as point identifiable). However, they are set identifiable [1], i.e., they can be identified up to equivalence classes consisting of parameters that result in the same probability density. Though different from point identifiability, set identifiability is nonetheless a useful property for likelihood-based generative models, as we are often more interested in the modeled density (and/or sampling) function(s) than the specific model parameters themselves. Therefore, we believe that MLE guarantees do have practical significance, in addition to their known theoretical properties.
>
> [1] The identification zoo: Meanings of identification in econometrics. https://www.aeaweb.org/articles?id=10.1257/jel.20181361
>
> [2] Unsupervised feature extraction by time-contrastive learning and nonlinear ICA. https://arxiv.org/abs/1605.06336
>
> [3] Ice-beem: Identifiable conditional energy-based deep models based on nonlinear ICA. https://proceedings.neurips.cc/paper/2020/hash/962e56a8a0b0420d87272a682bfd1e53-Abstract.html
>
> [4] On linear identifiability of learned representations. https://arxiv.org/abs/2007.00810

---

### Official Review · Reviewer_4m9N · 2022-11-02

**Confidence:** 5
**Correctness:** 2
**Technical Novelty And Significance:** 1
**Empirical Novelty And Significance:** 1
**Recommendation:** 3

**Clarity, Quality, Novelty And Reproducibility:**

The novelty is limited: the proposed idea is very similar to [1] except that the model is now a density model instead of an autoregressive score model. It also shares similarities with [2, 3]. The empirical performance is not very impressive compared to state-of-the-art DDPM and score-based models in terms of both likelihood and sample quality. The proposed evaluation metrics is not properly justified. The arguments and claims in section 3 should also be made more rigorous.

[1] Autoregressive Score Matching: https://arxiv.org/abs/2010.12810

[2] Improved Autoregressive Modeling with Distribution Smoothing: https://arxiv.org/abs/2103.15089

[3] Distribution Augmentation for Generative Modeling: http://proceedings.mlr.press/v119/jun20a/jun20a.pdf


**Strength And Weaknesses:**

Strength:
1. The paper considers a very interesting direction for autoregressive models.

2. This work has a decent amount of experiments and the performance of the proposed method is reasonable.

Weaknesses:
1. Limited novelty: training generative models on images perturbed with multiple noise levels is not a new idea. Similar ideas have been explored for diffusion models [1, 2], normalizing flows [3], and also autoregressive models [4, 5]. Specifically, the main idea is very similar to [4] except that the authors now use autoregressive models to parameterize a normalized density (which can be trained via maximum likelihood estimation), while [4] uses autoregressive models to parameterize "scores" and trained the models with score matching on images perturbed with various noise levels. The main idea is also similar to [5] (which is not discussed), except that in [5], only two noise levels are used. The conclusion that training autoregressive models on noisy images improves the sample quality and model robustness was also mentioned in [5]. Given these related works, the novelty of this work is questionable.

2. Some claims are flawed. For instance, in the Introduction, the claim that "diffusion models are poor likelihood models and cannot be trained via maximum likelihood" might not be correct. Diffusion models can be understood as hierarchical VAEs and can be trained by optimizing ELBO (which also optimizes likelihoods). Follow-up works [6, 7] also show that diffusion/score-based models can be good likelihood models.

3. Some related works are not discussed or compared. For instance, [5] is a closely related paper and should be discussed and compared. [8, 9] also consider conditioning on different data augmentations which share similarities with the proposed method. It would be good to also compare with these approaches.

4. The equations/notations are not defined properly in section 3.

5. Section 3.2 is not convincing. Only intuitions are provided without rigorous arguments or proof. It is unclear whether the arguments are correct or not.

6. It remains unclear why the noise-robustness experiments (section 4.1) are useful. The proposed sanity test for checking the robustness of likelihood models is not convincing or mathematically-justified It is unclear whether a better score implies better robustness. It would be more convincing to compare on some standard benchmarks for robustness.

7. The empirical performance is not very impressive. For instance,  [6, 7] are able to achieve much better performance.

[1] Denoising Diffusion Probabilistic Models: https://arxiv.org/abs/2006.11239

[2] Score-Based Generative Modeling through Stochastic Differential Equations: https://arxiv.org/abs/2011.13456

[3] Denoising Normalizing Flow: https://proceedings.neurips.cc/paper/2021/file/4c07fe24771249c343e70c32289c1192-Paper.pdf

[4] Improved Autoregressive Modeling with Distribution Smoothing: https://arxiv.org/abs/2103.15089

[5] Autoregressive Score Matching: https://arxiv.org/abs/2010.12810

[6] Variational Diffusion Models: https://arxiv.org/abs/2107.00630

[7] Maximum Likelihood Training of Score-Based Diffusion Models: https://arxiv.org/abs/2101.09258

[8] Distribution Augmentation for Generative Modeling: http://proceedings.mlr.press/v119/jun20a/jun20a.pdf

[9] Generating High Fidelity Images with Subscale Pixel Networks and Multidimensional Upscaling: https://arxiv.org/abs/1812.01608

**Summary Of The Paper:**

This work proposes a noise conditional likelihood-based method for training autoregressive models. Specifically, instead of directly training the model on clean images, the authors propose to train a sequence of autoregressive models on images perturbed with different noise levels using MLE. The authors show with empirical results that the proposed approach is able to improve the sample quality of autoregressive models while also achieving reasonably good likelihoods (in BPD).

**Summary Of The Review:**

Although this work considers a very interesting application of autoregressive models by incorporating noise-conditioning schemes from diffusion/score-based models, the technical novelty is limited and the paper presentation will also need to be improved.

---

> ### Author Response · Authors · 2022-11-09
> **Response to Reviewer 4m9N (3/3)**
>
> **Q: It would be more convincing to compare on some standard benchmarks for robustness.**
>
> **A:** We provide additional experimentation on anomaly and out-of-distribution (OOD) detection as further evidence for robustness. We use a benchmark initially conceived by [11] and extended by [5, 12, 13], where probabilistic models trained on CIFAR-10 are tasked to distinguish between in-distribution points (i.e., CIFAR-10 images) and out-of-distribution points (SVHN, constant value, and uniformly distributed images). We are happy to report that our noise conditional training greatly improves the robustness of our likelihood models, and we obtain SOTA results. Thank you for this suggestion, and have added this experiment to the text, along with further experimental details.
>
> | OOD Data / Model 	| PixelCNN++	| GLOW	| EBM	| AR-CSM	| NCPN (Ours)	|
> |---|---|---|---|---|---|
> | SVHN				| 0.32		| 0.24	| 0.63	| 0.68		| **0.74**			|
> | Const Uniform		| 0.0			| 0.0		| 0.30	| 0.57		| **0.68**			|
> | Uniform				| **1.0**			| **1.0**		| **1.0**		| 0.95		| **1.0**			|
> | Average			| 0.44		| 0.41	| 0.64	| 0.73		| **0.80**			|
>
> **Q: Denoising diffusion and score-based models (e.g., [6, 7]) achieve better performance.**
>
> **A:** This is not generally true. Diffusion models (e.g. [6, 7]) do achieve better performance on the CIFAR-10 dataset. However, on the ImageNet 64x64 dataset, our proposed model surpasses all other models in the literature in terms of average likelihood (including [6, 7]), setting a new state-of-the-art baseline (Table 1).
>
> **Q: More related works [4, 8, 9].**
> We discuss [8, 9] above. Reviewer Vbpt has also suggested [4]. We have added it to the text. Some differences:
> * The decomposition proposed by [4] introduces latent variables to the probabilistic model, which means that it can no longer evaluate the likelihood of the generative model. As a result, the authors resort to ELBO optimization for training. Our model retains likelihood evaluation and maximum likelihood estimation capabilities.
>  * [4] only considers a single denoising step for generation, which must be defined at training time. Our model can adaptively sample and denoise from multiple noise levels, depending on available computational resources.
> * The denoising step in [4] is autoregressive and requires 32x32=1024 evaluations of the underlying network on CIFAR-10. Conversely, each denoising step in our model requires only a single gradient evaluation of the network. Therefore our denoising process is ~50x faster than [4].
>  * We achieve much better sample quality and density estimation performance in terms of FID and BPD.
>
> [1] Denoising Diffusion Probabilistic Models. https://arxiv.org/abs/2006.11239
>
> [2] Score-Based Generative Modeling through Stochastic Differential Equations. https://arxiv.org/abs/2011.13456
>
> [3] Denoising Normalizing Flow. https://proceedings.neurips.cc/paper/2021/file/4c07fe24771249c343e70c32289c1192-Paper.pdf
>
> [4] Improved Autoregressive Modeling with Distribution Smoothing. https://arxiv.org/abs/2103.15089
>
> [5] Autoregressive Score Matching. https://arxiv.org/abs/2010.12810
>
> [6] Variational Diffusion Models. https://arxiv.org/abs/2107.00630
>
> [7] Maximum Likelihood Training of Score-Based Diffusion Models. https://arxiv.org/abs/2101.09258
>
> [8] Distribution Augmentation for Generative Modeling. http://proceedings.mlr.press/v119/jun20a/jun20a.pdf
>
> [9] Generating High Fidelity Images with Subscale Pixel Networks and Multidimensional Upscaling. https://arxiv.org/abs/1812.01608
>
> [10] Deep Unsupervised Learning using Nonequilibrium Thermodynamics. https://arxiv.org/abs/1503.03585
>
> [11] Do Deep Generative Models Know What They Don't Know? https://arxiv.org/abs/1810.09136
>
> [12] Deep Anomaly Detection by Outlier Exposure. https://arxiv.org/abs/1812.04606
>
> [13] Implicit Generation and Generalization in Energy-Based Models. https://arxiv.org/abs/1903.08689
>
> [14] Scheduled Sampling for Sequence Prediction with Recurrent Neural Networks. https://arxiv.org/abs/1506.03099
>
> [15] Professor Forcing: A New Algorithm for Training Recurrent Networks. https://arxiv.org/abs/1610.09038

---

> ### Author Response · Authors · 2022-11-09
> **Response to Reviewer 4m9N (2/3)**
>
> **Q: Some claims are flawed: Diffusion/score-based models can be good likelihood models [6, 7].**
>
>  **A:** Indeed, diffusion models can attain good likelihoods as shown by [6, 7]. However, these evaluations are expensive, as diffusion models have no closed form for likelihoods, and require hundreds to thousands of number of function evaluations (NFEs) to evaluate a singe likelihood, whereas autoregressive (and normalizing flow-based) models require a single NFE. Moreover, as stated above, diffusion models cannot be trained by MLE, and therefore do not enjoy the same asymptotic guarantees. In this regard, diffusion models are inferior to autoregressive (and flow-based) models. However, we agree that we may have overstated our point, and it could easily be taken out of context. We have removed the statement: "However, [diffusion models] are poor likelihood models..." from the text.
>
> **Q: Other related works: [5, 8, 9].**
>
> **A:** We have reviewed [9], and did not find any mention of data augmentation in the training process. Moreover, we do discuss [5] in Section 2.2.
>
> [8] indeed appears to be relevant to our work, and also considers data augmentations for autoregressive generative modeling. We have added it to the text. Some key differences and observations:
> * We consider a single type of noise-based augmentation, which as you have already mentioned, is a well-known and principled approach to augmenting generative models. On the other hand, [8] uses a set of heuristically chosen and experimentally pruned augmentations, that include RandAugment, color swapping, crop + translations, and jigsaw shuffling.
>  * [6] appears to also use the same set of augmentations as [8] to further improve their CIFAR-10 diffusion model from 2.65 BPD to 2.49 BPD. Of course, diffusion models by default already use our noise conditional augmentation. This suggests that [8] complements, rather than competes with, our work. We thus believe that the addition of their augmentations will further improve modeling performance.
> * We achieve better sample quality with smaller models (FID: 12.09 vs 12.75, parameters: 78M vs 152M).
>
> **Q: The equations/notations are not properly defined in Section 3. Section 3.2 requires more rigor.**
>
> **A:** We have greatly improved the clarity and rigor of Section 3 by reformatting 3.1 and formalizing several of the arguments in 3.2. More concretely, in 3.1, we provided a cleaner form for $p_\pi$, which does involve a slight adjustment to its definition &mdash; we have updated the relevant experiments, but they have minimally changed. In 3.2, we have added Lemma 3.1, which formally proves that $q_{human}$ and $p_\theta$ are different under the observed conditions, and Lemma 3.2, which connects the expected likelihood under $p_\pi$ to a marginalized likelihood under $p_{data}$, where the least significant bit (LSB) is marginalized out, thus showing more rigorously that prior gains center on modeling the LSB.
>
> **Q: It remains unclear why the noise-robustness experiments (section 4.1) are useful. The proposed sanity test for checking the robustness of likelihood models is not convincing or mathematically-justified.**
>
>  **A:** Unstable sequential sampling due to compounding sampling errors is well-documented in autoregressive generative models [4, 14, 15] (also see Figure 1). The noise-robustness experiments serve to quantify robustness to such errors. The proposed sanity test can be seen as measuring the robustness to errors in the least significant bit (LSB). (See Lemma 3.2.) In 8-bit images, the LSB contains image features that are almost entirely visually inconsequential (Figure 2), and perturbing it constitutes the smallest possible error that can be effected on an image. **Therefore, the proposed sanity test serves two purposes**: First, it provides a necessary (but not sufficient) condition for stable sampling. Second, it acts as a measure for modeling performance on the non-LSB (and thus more visually important features) in an image.

---

> ### Author Response · Authors · 2022-11-11
> **Response to Reviewer 4m9N (1/3)**
>
> Thank you for your thorough review. Some of your concerns seem to be misguided, and we hope we addressed the confusions. We have responded to each of your comments below. Moreover, we've incorporated your feedback into a revision of the paper, and believe it has notably improved the readability and persuasiveness our text.
>
> **Q: Limited Novelty: Training generative models on images perturbed with multiple noise levels is not a new idea (see [1, 2, 3, 4, 5]).**
>
> **A:** We agree with this statement, but also argue that this is too reductionist a view in terms of assessing novelty in this area. By your interpretation, all related works you cite [1-7] are also not novel. (See, for example, [10], a well-known work which predates [1-7], and similarly proposes a probabilistic generative model trained on a diffusion process applied to the data.)
>
> Therefore, we argue that a fruitful discussion of novelty will require more nuance. We emphasize that our novelty lies **not** in simply training with noisy data --- this is in fact the starting point for our research: We sought to improve maximum likelihood estimation with ideas inspired by denoising / diffusion models. This core principle --- **denoising models trained via maximum likelihood estimation, and that inherit all the guarantees thereof, is what separates our work from all other mentioned works.** For example, [1, 5] are trained by optimizing a score matching loss, and [2, 4, 6, 7] by a variational lower bound on the likelihood. None of these schemes produce a density model that enjoys the theoretical guarantees of maximum likelihood estimation. Perhaps most related to our work is [3], which involves regularized maximum likelihood estimation, i.e. optimizing the average likelihood plus a non-likelihood regularization term. However, the model is not applied to realistic data, so its practical performance is unclear. Furthermore, its regularization term breaks the aforementioned maximum likelihood guarantees. Finally, we note that [5] is not actually trained on multiple noise levels, contrary to what is suggested. Though we firmly stand by the novelty of our work, we greatly appreciate your thorough review of the literature, and have incorporated the relevant parts of it into the related works section of our text.
>
> **Q: Some claims are flawed: Diffusion models can be understood as hierarchical VAEs and can be trained by optimizing ELBO (which also optimizes likelihoods).**
>
> **A:** While this statement is correct, its implication, that diffusion models also perform maximum likelihood estimation, is not. Maximum likelihood estimation is not simply any optimization process that increases the model likelihood. Rather, it is a specific technique for parameter estimation via direct maximization of the likelihood. While ELBO maximization also increases the likelihood via a lower bound, it provides no asymptotic guarantees on the correctness of the resulting model, for general parametric models. Moreover, maxima for ELBO may not even coincide with maxima for the model log likelihood. **In other words, solutions that maximize ELBO are not necessarily solutions of maximum likelihood**. Therefore, diffusion models (which are trained with denoising score matching or ELBO optimization) are theoretically distinct from our model. We do see how this confusion can occur, as this distinction is further blurred by a recent and influential paper [7], which describes ELBO maximization as maximum likelihood.
>
> To help reduce this confusion, we have changed all instances in the paper of "maximum likelihood" and "likelihood maximization models" to "maximum likelihood estimation" and "MLE-based models", so as to more explicitly highlight this distinction. We have also emphasized this distinction in Sections 1 and 2.

---

### Author Response · Authors · 2022-11-09
**Summary of Responses to Reviewers (2/2)**

**3) Formalized Arguments in Section 3.** Under the suggestion of Reviewer 4m9N, we cleaned up the definition of $p_\pi$ and formalized several of the arguments in 3.2. Of greatest interest is Lemma 3.2. Reviewers 4m9N and Vbpt question the utility of the proposed sanity test, which involves evaluating the model likelihood over a noise-perturbed distribution $p_\pi$. Lemma 3.2 connects the expected likelihood under $p_\pi$ to an expected marginal likelihood under $p_{data}$, where the least significant bit (LSB) is marginalized out, thus showing more rigorously that that the bit-perturbation experiments measure modeling performance on the remaining $k-1$ most significant bits, and highlighting that prior gains in likelihood models center on modeling the LSB, while NCML trained models distribute modeling capacity more evenly across bits.

---

### Author Response · Authors · 2022-11-11
**Summary of Responses to Reviewers (1/2)**

We thank all reviewers for their feedback. Overall, reviewers found the work interesting (4m9N, 4nNY, YYAi), well written (YYAi), and that it meaningfully improved the performance of autoregressive models (4m9N, Vbpt, 4nNY, YYAi). However, they were split on their final decision. We firmly believe that most of the objections raised are due to simple misunderstandings of our motivation and proposed model, rather than fundamental flaws in the research.

We have addressed each reviewer's concerns separately and summarized the main points below. Moreover, we have heavily incorporated reviewer feedback into a revision of the paper. Major changes are highlighted. We believe that the exposition and empirical insights of our paper have significantly improved as a result.

**1) Maximum Likelihood Estimation (MLE) versus other methods for training generative models (e.g. denoising score matching, variational inference).**

Our work is motivated by improving MLE in generative modeling using recent ideas in diffusion models, while preserving the theoretical guarantees of MLE. Reviewers 4m9N and Vbpt raised concerns over this motivation, stating that diffusion models already enjoy similar guarantees as MLE trained models. The two arguments were that **1) score matching also provides some of the same guarantees of MLE** (Vbpt) and **2) ELBO optimization also increases likelihoods** (4m9N, Vbpt). While both statements are true, they do not imply that diffusion models have MLE-like asymptotic guarantees. **For statement 1)**, this is because score matching itself does not produce a diffusion model. After score matching training, one must additionally define and integrate an SDE involving the scores to obtain the diffusion model. This choice of SDE is very important, yet not at all dictated by score matching. Thus, while each learned score enjoys asymptotic guarantees (i.e., consistency, asymptotic normality), the diffusion model has no known guarantees. **For statement 2)**, we note that MLE is not simply any training process that increases the likelihood. While the ELBO is a lower bound on the likelihood and thus provably increases the likelihood over training, the minima of the ELBO and likelihood are not guaranteed to be the same. For this reason, there are also no known asymptotic guarantees conferred by ELBO optimization for general neural network models. **Ultimately, we maintain that all competing models are either inferior in performance (both in terms of sample quality and density estimation), or do not provide any of the same theoretical guarantees as MLE (or both).** Moreover, we have equal or better likelihoods on ImageNet 64x64 than all known models.

**2) Further empirical validation of the robustness properties of NCML.**

One of our core contributions is improved robustness to perturbations in the input. A general sentiment was that it would be illuminating to see further experiments validating the utility of this robustness, outside of improving sample quality (Vbpt, 4m9N, 4nNY). 4m9N expressed interest in measuring robustness with a more standard benchmark. Vbpt noted that, while we discussed downstream applications that involve the likelihood, we did not include any such functionality in our work. 4nNY expressed concern about whether the added noise robustness would resolve known issues with likelihoods discussed in literature. To this end, we have added further experimentation. We use a task originally conceived to highlight known issues of likelihood models in producing semantically meaningful likelihoods, which was subsequently extended into an out-of-distribution (OOD) detection benchmark. In brief, models trained on CIFAR-10 are tasked to discriminate in-distribution (i.e., CIFAR-10) images from OOD (i.e., SVHN, uniform gray, and random uniform) images. We show that our noise-robust models not only outperform the MLE baselines (PixelCNN++, GLOW) which are known to fail this task, but even SOTA non-MLE models (AR-CSM and EBM).
| 	| PixelCNN++	| GLOW	| EBM	| AR-CSM	| NCPN (Ours)	|
|---|---|---|---|---|---|
| SVHN				| 0.32		| 0.24	| 0.63	| 0.68		| **0.74**			|
| Const Uniform		| 0.0			| 0.0		| 0.30	| 0.57		| **0.68**			|
| Uniform				| **1.0**			| **1.0**		| **1.0**		| 0.95		| **1.0**			|
| Average			| 0.44		| 0.41	| 0.64	| 0.73		| **0.80**			|

We additionally compare bit perturbation robustness to performance on the OOD detection task, and find that improved noise-robustness by our proposed test correlates well with performance on this OOD detection task.

| Task / Model 		| MLE		| NCML (VE)	| NCML (sub-VP)	| NCML (VP)	|
|---|---|---|---|---|
| SVHN			| 0.35		| 0.43		| 0.65			| **0.74**		|
| Const Uniform	| 0.1			| 0.56		| 0.58			| **0.68**		|
| Uniform		| **1.0**	| **1.0**	| **1.0**		| **1.0**			|
| Average		| 0.48		| 0.66		| 0.74			| **0.80**		|
| BPD: CIFAR ($\pi = 1$)	| 3.99		| 3.94		| 3.93			| **3.89**		|

---

### Decision · Program_Chairs · 2023-01-20

**Decision:**

Reject

**Justification For Why Not Higher Score:**

The lack of novelty of the main method and lack of clarity in some important parts of the paper.

**Justification For Why Not Lower Score:**

N/A

**Metareview: Summary, Strengths And Weaknesses:**

**Summary:**
The paper introduces noise conditional likelihood-based methods for training autoregressive models. The authors argue that standard likelihood approaches are sensitive to imperceptible perturbations which degrades their sampling quality. The proposed method aims to address this by training a model not only on training data but also on noisy versions of it while conditioning the model on the noise level. The authors also propose a sampling procedure that anneals between a noisy model and a non-noisy one using Langevin-like dynamics which exploits the score of the learned density model. Empirical findings illustrate an increased sample quality and robustness to small perturbations.

**Summary of the discussions:**

Reviewers YYAi and 4nNY gave very positive reviews highlighting the novelty in the noise conditional likelihood-based method and the motivation for learning models robust to small perturbations of the data. I personally found interesting the fact that conditioning on noise improves likelihood training much like in denoising score matching.

On the other hand, reviewers Vbpt and 4m9N provided negative reviews with the main criticism being the lack of novelty and clarity. In particular, it appears that the idea of conditioning the likelihood model on noise appeared in a paper [1]. There, the model is conditioned on augmentations, which can include gaussian noise parameterized by its variance. The proposed objective in the submission becomes a particular case of the one in [1].

The discussion about robustness to LSB would also benefit from clarity as pointed out by Vbpt and YYAi, in particular, to what extent the conclusions in discrete space extend to a continuous setting (which is the setting of interest) and how the proposed sanity test motivates the proposed method. For instance, Figure 4 shows that the test decreases gradually when conditioning the model on larger noise (parameterized by t). However, as t increases, the learned model (conditional on t) looks like a gaussian. Hence the likelihood of such a model (at time t) under the data amounts to computing the second moment of the data. Therefore, the test only suggests that data distributions and its LSB perturbation have similar second moments (for large t).

Although the authors provided clarifications and added more experiments to illustrate the benefits of the method on downstream tasks, the current response does not adequately address the novelty concerns and many claims about alternative methods (such as diffusion processes, denoising score-matching) are not well supported and too dismissive without discussion the (obvious and less obvious) benefits they could have.

**Conclusion:**
Overall, the paper would benefit from additional work to clarify the contributions and contextualize existing works. Putting more emphasis on the sampling part and downstream applications could also be beneficial.

**Reference**
[1] Distribution Augmentation for Generative Modeling: http://proceedings.mlr.press/v119/jun20a/jun20a.pdf


**Summary Of Ac-Reviewer Meeting:**

Please refer to the main meta-review for a summary of the discussion.

The lack of novelty and clarity are the main reasons for rejection as this would require substantial changes to the paper to adequately address those concerns.